# Learning developmental mode dynamics from single-cell trajectories

**Nicolas Romeo[1,2†], Alasdair Hastewell[1†], Alexander Mietke[1*], Jörn Dunkel[1*]**

[1]Department of Mathematics, Massachusetts Institute of Technology, Cambridge, United States; [2]Department of Physics, Massachusetts Institute of Technology, Cambridge, United States

**Abstract** Embryogenesis is a multiscale process during which developmental symmetry breaking transitions give rise to complex multicellular organisms. Recent advances in high-resolution live-cell microscopy provide unprecedented insights into the collective cell dynamics at various stages of embryonic development. This rapid experimental progress poses the theoretical challenge of translating high-dimensional imaging data into predictive low-dimensional models that capture the essential ordering principles governing developmental cell migration in complex geometries. Here, we combine mode decomposition ideas that have proved successful in condensed matter physics and turbulence theory with recent advances in sparse dynamical systems inference to realize a computational framework for learning quantitative continuum models from single-cell imaging data. Considering pan-embryo cell migration during early gastrulation in zebrafish as a widely studied example, we show how cell trajectory data on a curved surface can be coarse-grained and compressed with suitable harmonic basis functions. The resulting low-dimensional representation of the collective cell dynamics enables a compact characterization of developmental symmetry breaking and the direct inference of an interpretable hydrodynamic model, which reveals similarities between pan-embryo cell migration and active Brownian particle dynamics on curved surfaces. Due to its generic conceptual foundation, we expect that mode-based model learning can help advance the quantitative biophysical understanding of a wide range of developmental structure formation processes.

**\*For correspondence:**
amietke@mit.edu (AM);
dunkel@math.mit.edu (JD)

[†]These authors contributed equally to this work

**Competing interest:** The authors declare that no competing interests exist.

## Editor's evaluation

This work proposes a method to obtain a reduced description of the collective dynamics of thousands of cells moving together during zebrafish gastrulation as a few fundamental modes, and to derive effective dynamics for these modes. This well-written work enables a simplified picture of the key features of cellular collective motion, that will be useful to physicists and biologists looking for a quantitative understanding of morphogenesis.

## Introduction

Embryogenesis, the development of a multicellular organism from a single fertilized egg cell, requires coordinated collective motions of thousands of cells across a wide range of length and time scales (*Gilbert and Barresi, 2016*; *Solnica-Krezel, 2005*). Understanding how a highly reproducible and robust tissue organization arises from the dynamics and interactions of individual cells presents a major interdisciplinary challenge (*Collinet and Lecuit, 2021*). Recent advances in high-resolution live imaging make it possible to track the internal biological states and physical movements of many individual cells on pan-embryonic scales throughout various stages of development (*Stelzer, 2015*; *Power and Huisken, 2017*; *Hartmann et al., 2019*; *Shah et al., 2019*). This unprecedented wealth of data poses two intertwined compression problems of equal practical and conceptual importance.

The first concerns the efficient reduction of high-dimensional tracking data without loss of relevant information; the second relates to inferring predictive low-dimensional models for the developmental dynamics. Mathematical solutions to the first problem are aided by taking into account the geometry and symmetries of the developing embryo, which suggest suitable basis functions for a coarse-grained and sparse mode representation of raw data (*Levy, 2006*). Efficient algorithmic approaches tackling the second problem appear within reach thanks to recent advances in the direct inference of dynamical systems equations from data (*Brunton et al., 2016*; *Rackauckas et al., 2021*). Building on these ideas, we construct and demonstrate here a computational framework that translates developmental single-cell trajectory data on curved surfaces into quantitative models for the dominant hydrodynamic modes.

Widely applied in physics (*Kac, 1966*; *Goldenfeld and Woese, 2011*; *Kantsler and Goldstein, 2012*; *Bhaduri et al., 2020*), engineering (*Soong and Grigoriu, 1993*; *Heydari et al., 2021*), and spectral computing (*Driscoll et al., 2014*; *Burns et al., 2020*; *Fortunato et al., 2021*), mode representations (*Schmid, 2010*; *Tu et al., 2014*) provide a powerful tool to decompose and study system dynamics at and across different energetic, spatial and temporal scales. In quantum systems, for example, mode representations in the form of carefully constructed eigenstates are used to characterize essential energetic system properties (*Slater and Koster, 1954*; *Jaynes and Cummings, 1963*). Similarly, turbulence theory has seen significant progress by studying the coupling between Fourier modes that represent dynamics at different length scales. This approach enabled a better understanding of energy cascades (*Kolmogorov, 1941*; *Wang et al., 2021*) and provided insights into the nature of turbulence in non-living (*Kraichnan and Montgomery, 1980*; *Pope, 2000*) and in living (*Dunkel et al., 2013*; *Bratanov et al., 2015*; *Ramaswamy and Jülicher, 2016*; *Alert et al., 2020*) systems. Additionally, the multi-scale nature of many biological processes make them particularly amenable to a representation in terms of spatial and temporal modes (*Marchetti et al., 2013*). Despite this fact, however, mode representations are not yet widely used to characterize and compress cell tracking data, or to infer dynamic models from such data.

To demonstrate the practical potential of mode representations for the description of multicellular developmental processes, we develop here a computational framework that takes cell tracking data as inputs, translates these data into a sparse mode representation by exploiting symmetries of the biological system, and utilizes recently developed ODE inference techniques (*Rackauckas et al., 2021*) to infer a predictive dynamical model. The model will be specified in terms of a learned Green's function that propagates initial cell density and flux data forward in time. To validate the approach, we demonstrate that it correctly recovers the hydrodynamic equations for active Brownian particle (ABP) dynamics on curved surfaces. Subsequently, as a first example application to experimental single-cell tracking data, we consider the pan-embryonic cell migration during early gastrulation in zebrafish (*Shah et al., 2019*), an important vertebrate model system for studying various morphogenetic events (*Solnica-Krezel, 2005*; *Krieg et al., 2008*; *Morita et al., 2017*). During gastrulation, complex migratory cell movements organize several thousand initially undifferentiated cells into different germlayers that lay out the primary body plan (*Rohde and Heisenberg, 2007*). The underlying high-dimensional single-cell data make this process a prototypical test problem for illustrating how spatio-temporal information can be efficiently compressed to analyze and model biological structure formation.

## Results

Broadly, our goal is to translate experimentally measured single-cell trajectories on a curved surface into a quantitative model of collective cell migration dynamics. As a specific example, we consider recently published lightsheet microscopy data that captures the individual movements of thousands of cells during early zebrafish development from epiboly onset at 4 hours post-fertilization (hpf) to about 18 hpf (*Shah et al., 2019*). This developmental period is characterized by a collective symmetry breaking event during which cells collectively migrate over the yolk cell surface (*Rohde and Heisenberg, 2007*). Namely, they rearrange from an initial localization around the animal pole (AP) (*Figure 1A*, left) into a more elongated configuration that already indicates the basic geometry of the fully developed zebrafish larva (*Figure 1A*, right). Working with a two-dimensional (2D) sphere projection of the experimental data, we first describe a coarse-graining approach that faithfully captures cell-mass transport on a curved surface. We then construct a sparse mode representation of the resulting hydrodynamic fields in terms of scalar and vector spherical harmonic basis functions, discuss

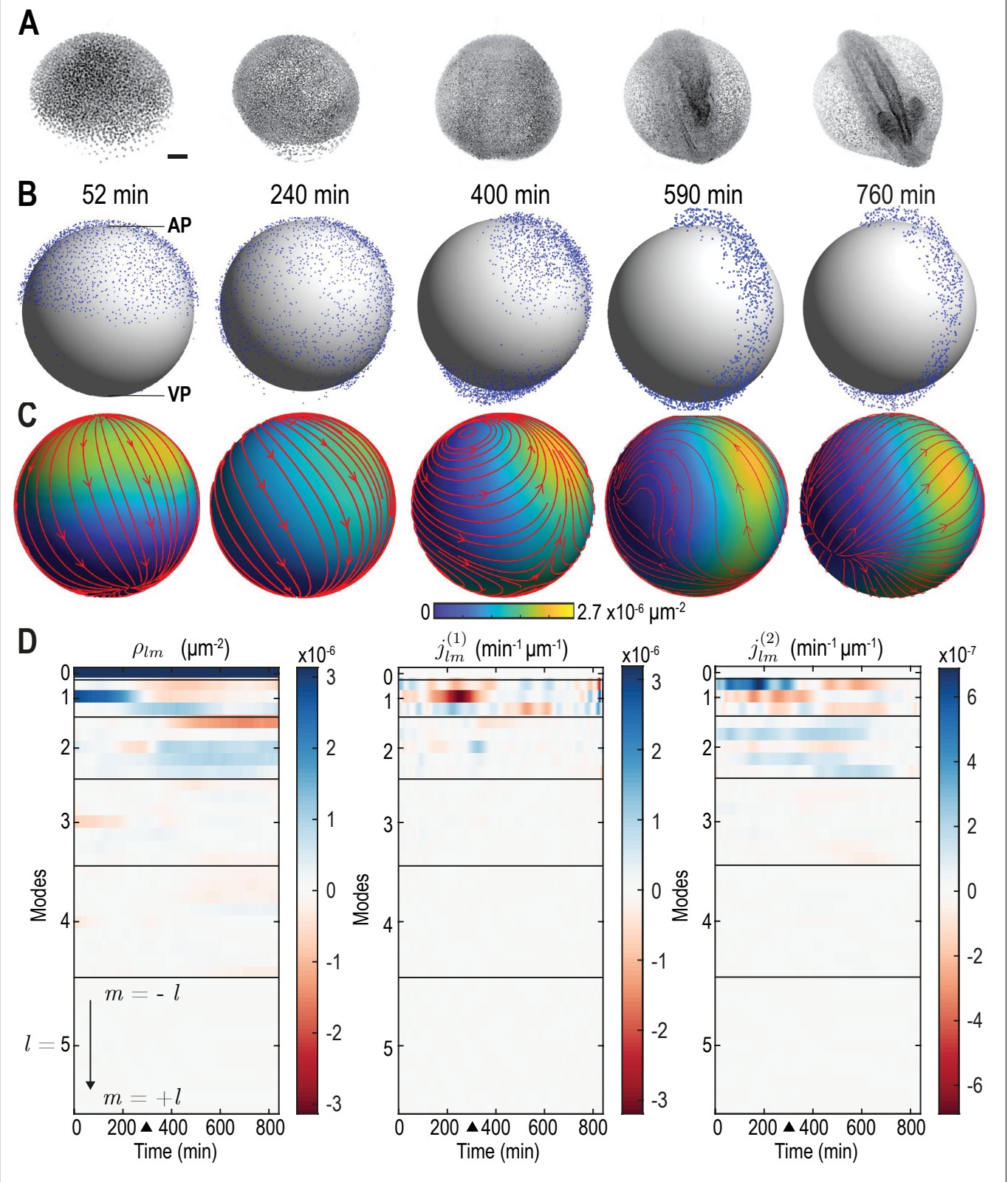

**Figure 1.** From single-cell tracking data to sparse mode amplitude representations. (**A**) Microscopic imaging data of early zebrafish development (adapted from Figure 1b in *Kobitski et al., 2015*) shows cell migration from an initially homogeneous pole of cells (left) toward an elongated structure that indicates the head-tail axis of the fully developed organism. Scale bar, $100\,\mu\text{m}$. (**B**) Experimental single-cell tracking data from *Shah et al., 2019* (blue dots) during similar developmental time points ($\pm20$ min) as in A. $t = 0$ min for the indicated time points in B corresponds to a developmental time

*Figure 1 continued on next page*

*Figure 1 continued*

of 4 hours post fertilization. The $z$-axis points from the ventral pole (VP) to the animal pole (AP). (**C**) Coarse-grained relative cell density $\rho(\mathbf{r}, t)$ (color) and associated coarse-grained flux $\mathbf{J}(\mathbf{r}, t)$ (streamlines) determined from single cell positions and velocities from data in B via *Equation 2a*. Thickness of streamlines is proportional to the logarithm of the spatial average of |$\mathbf{J}$| (see *Video 1*). (**D**) Dynamic harmonic mode representation of the relative density $\rho(\mathbf{r}, t)$ (*Equation 4*, left panel) and of the flux $\mathbf{J}(\mathbf{r}, t)$ (*Equation 5*, middle and right panel) for fields shown in C. The modes $j_{lm}^{(1)}$ correspond to compressible, divergent cell motion, the modes $j_{lm}^{(2)}$ describe incompressible, rotational cell motion. Mode amplitudes become negligible for $l \geq 5$ (*Video 2*). For all panels, horizontal black lines delineate blocks of constant harmonic mode number $l$ and black triangles denote the end of epiboly phase.

The online version of this article includes the following figure supplement(s) for figure 1:

**Figure supplement 1.** Convergence of spectral representation.

mode signatures of morphogenetic symmetry breaking events, and connect them to the dynamics of topological defects in the cellular flux. We validate this mode representation framework and the subsequent model inference using synthetic data of ABPs on a sphere, for which coarse-grained fields and learned models can be directly compared against analytical predictions. Finally, we infer a linear model for the mode dynamics of the experimental zebrafish data, which enables us to study the characteristics of cell interactions through kernels that couple cell density and flux and compare their features with the hydrodynamic mean-field signatures of ABPs on a sphere.

## Coarse-graining of cellular dynamics on a spherical surface

The experimentally observed cell motions are approximately two-dimensional (2D): The radius of the yolk cell surface on which the dynamics takes place is much larger than the average height changes of the evolving cell mass (*Shah et al., 2019*). We therefore adopt a thin film approximation, in which the cellular motion is represented on an effective spherical mid-surface (gray surface in *Figure 1B*); refined future models should aim to account for the full 3D dynamics. Focusing here on the in-plane dynamics, we project all cell positions and velocities onto a spherical mid-surface $\mathcal{S}$ of radius $R_s = 300\,\mu$m. On this spherical surface, each cell $\alpha = 1, 2, ..., N$ has a position $\mathbf{r}_\alpha(t)$ and in-plane velocity $\mathbf{v}_\alpha(t) = \mathrm{d}\mathbf{r}_\alpha/\mathrm{d}t$.

As a second processing step, a coarse-grained representation of the single-cell dynamics on a spherical surface is determined. To facilitate the applicability of our framework to a wide range of experimental inputs, we propose a coarse-graining approach that can flexibly integrate cell number variations stemming from cell divisions, but also those from experimental uncertainties in cell imaging and tracking. Consequently, we first consider an idealized scenario in which the total cell number is approximately constant. In this case, mass conservation informs the construction of self-consistent coarse-graining kernels on a spherical surface. In a second step, we describe how this approach generalizes when there are variations in the total cell number.

### Consistent coarse-graining of idealized microscopic data

Our specific aim is to translate microscopic cell positions $\mathbf{r}_\alpha(t)$ and velocities $\mathbf{v}_\alpha(t)$ into a continuous cell surface density $\rho(\mathbf{r}, t)$ and an associated flux $\mathbf{J}(\mathbf{r}, t)$ at any point $\mathbf{r}$ of the spherical mid-surface. For an approximately constant total number of cells, the fields $\rho$ and $\mathbf{J}$ are related by the mass conservation equation

**Learning developmental mode dynamics from single-cell trajectories**

Nicolas Romeo, Alasdair D. [...]xander Mietke, Jörn Dunkel

**Movie S1**

**Video 1.** Time evolution of the pre-processed cell tracking data (point cloud, see Materials and methods), and of the density field $\rho(\mathbf{r}, t)$ (colormap) and associated flux $\mathbf{J}(\mathbf{r}, t)$ (streamlines) corresponding to the harmonic modes $\{\rho_{lm}, j_{lm}^{(1)}, j_{lm}^{(2)}\}$ shown in Figure 1D. This mode representation was determined by the coarse-graining and projection procedure described in the main text. Streamline thickness is proportional to the logarithm of the average flux amplitude $\langle|\mathbf{J}|\rangle_s$. For visualization purposes, cell distances to the origin were rescaled by a factor of $1.2R_s/\langle R(t)\rangle$, where $\langle R(t)\rangle$ is the average cell distance from center at time $t$ and $R_s = 300\,\mu$m is the mid-surface radius.

https://elifesciences.org/articles/68679/figures#video1

$$\frac{\partial \rho}{\partial t} + \nabla_{\mathcal{S}} \cdot \mathbf{J} = 0. \tag{1}$$

Here, $\nabla_{\mathcal{S}} \cdot \mathbf{J}$ denotes the in-plane divergence of the cell number flux. To convert cell position $\mathbf{r}_\alpha(t)$ and velocities $\mathbf{v}_\alpha(t)$ into a normalized cell surface density $\rho(\mathbf{r}, t)$ and an associated normalized flux $\mathbf{J}(\mathbf{r}, t)$, we consider a kernel coarse-graining of the form (Appendix 1)

$$\rho(\mathbf{r}, t) = \frac{1}{N} \sum_{\alpha=1}^{N} K\left[\mathbf{r}, \mathbf{r}_\alpha(t)\right] \tag{2a}$$

$$\mathbf{J}(\mathbf{r}, t) = \frac{1}{N} \sum_{\alpha=1}^{N} \mathcal{K}\left[\mathbf{r}, \mathbf{r}_\alpha(t)\right] \cdot \bar{\mathbf{v}}_\alpha, \tag{2b}$$

where $N$ is the total number of cells and $\bar{\mathbf{v}}_\alpha = \mathbf{v}_\alpha/|\mathbf{r}_\alpha|$ is the angular velocity of a given cell on a reference unit sphere (Appendix 1). The kernels $K(\mathbf{r}, \mathbf{r}')$ and $\mathcal{K}(\mathbf{r}, \mathbf{r}')$ are given by a scalar and a matrix-valued function, respectively. The matrix kernel $\mathcal{K}(\mathbf{r}, \mathbf{r}')$ takes into account contributions of a particle with velocity $\mathbf{v}_\alpha$ at $\mathbf{r}' = \mathbf{r}_\alpha$ to nearby points $\mathbf{r}$ on the sphere, which involves an additional projection to ensure that $\mathbf{J}(\mathbf{r}, t)$ is everywhere tangent to the spherical surface (Appendix 1). Importantly, the mass conservation *Equation 1* implies a non-trivial consistency relation between the kernels $K(\mathbf{r}, \mathbf{r}')$ and $\mathcal{K}(\mathbf{r}, \mathbf{r}')$ in *Equation 2a*, *Equation 2b*. The kernels that obey this condition represent different coarse-graining length scales (*Appendix 1—figure 2*). Throughout, we fix an intermediate coarse-graining length scale to enable a sparse representation of the experimental data, while ensuring that spatial details of the dynamics remain sufficiently well resolved. The final surface density $\rho(\mathbf{r}, t)$ and the associated normalized flux $\mathbf{J}(\mathbf{r}, t)$, computed from *Equation 2a* and *Equation 2b* using a kernel with an effective great-circle coarse-graining width of $\sim 70\,\mu\text{m}$, are shown in *Figure 1C* (see also *Video 1*).

## Consequences of cell number variations in experimental data

Because cell divisions are essential to most developmental processes, total cell numbers will in many cases – including early zebrafish gastrulation (*Kobitski et al., 2015*) – vary over time. True cell numbers and cell number changes are often difficult to measure due to experimental uncertainties arising from single-cell imaging and tracking within dense cellular aggregates. We therefore merely assume here that single cells are tracked in a representative fashion so that local relative surface densities found from *Equation 2a* reflect the probability that cells are present at a given point $\mathbf{r}$. In the absence of further information on cell deaths and cell divisions, we additionally make the more restrictive assumption that cell appearances or disappearances are everywhere proportional to the local cell density. With these assumptions, we can define a cell number surface density $\tilde{\rho}(\mathbf{r}, t) = N(t)\rho(\mathbf{r}, t)$, where $N(t)$ is the cell number at time $t$ and $\rho(\mathbf{r}, t)$ is the normalized surface density given in *Equation 2a*. Similarly, a cell number flux is given by $\tilde{\mathbf{J}}(\mathbf{r}, t) = N(t)\mathbf{J}(\mathbf{r}, t)$, where the flux $\mathbf{J}(\mathbf{r}, t)$ is computed from the data as described by *Equation 2b*. Using these definitions in *Equation 1*, we find that the fields $\tilde{\rho}(\mathbf{r}, t)$ and $\tilde{\mathbf{J}}(\mathbf{r}, t)$ obey a continuity equation

$$\frac{\partial \tilde{\rho}}{\partial t} + \nabla_{\mathcal{S}} \cdot \tilde{\mathbf{J}} = k(t)\tilde{\rho}, \tag{3}$$

where $k(t) = \dot{N}(t)/N(t)$ denotes a time-dependent effective growth rate. Importantly, under the two above assumptions, *Equation 3* encodes for any time-dependent total cell number $N(t) > 0$ the same information as *Equation 1* for coarse-grained normalized surface density $\rho(\mathbf{r}, t)$ and associated flux $\mathbf{J}(\mathbf{r}, t)$ given by *Equation 2a* and *Equation 2b*, respectively. In the following analysis, we hence focus on these normalized fields.

## Spatial mode representation on a spherical surface

To obtain a sparse mode representation of the hydrodynamic fields $\rho(\mathbf{r}, t)$ and $\mathbf{J}(\mathbf{r}, t)$ on the spherical surface, we expand them in terms of scalar and vector spherical harmonics (SHs) (*Arfken et al., 2013*; *Sandberg, 1978*) (Appendix 2.A). SHs are defined on points $\hat{\mathbf{r}} = \mathbf{r}/R_s$ of the unit sphere, where $R_s = 300\,\mu\text{m}$ is the mid-surface radius. In this basis, the scalar density field is represented as

$$\rho(\mathbf{r}, t) = \sum_{l=0}^{l_{\max}} \sum_{m=-l}^{l} \rho_{lm}(t) Y_{lm}(\hat{\mathbf{r}}), \tag{4}$$

Learning developmental mode dynamics
from single-cell trajectories

Nicolas Romeo, Alasdair D. ████xander Mietke, Jörn Dunkel

Movie S2

**Video 2.** Reconstruction of the hydrodynamics fields in real space by adding consecutive scalar and vector spherical harmonic modes of progressively higher order $l$. Surface coloring depicts the density field $\rho(\mathbf{r}, t)$, the associated flux $\mathbf{J}(\mathbf{r}, t)$ is indicated by streamlines. Streamline thickness is proportional to the logarithm of the average flux amplitude $\langle |\mathbf{J}| \rangle_s$. The shown fields correspond to the time point $t = 420$ min in Video 1.
https://elifesciences.org/articles/68679/figures#video2

which conveniently separates the time- and space-dependence of $\rho(\mathbf{r}, t)$ into mode amplitudes $\rho_{lm}(t)$ and scalar harmonic functions $Y_{lm}(\hat{\mathbf{r}})$, respectively. The maximal mode number $l_{\max}$ is a proxy for the maximal spatial resolution at which $\rho(\mathbf{r}, t)$ is faithfully represented. Similarly, the vector-valued flux $\mathbf{J}(\mathbf{r}, t)$ can be decomposed into time-dependent mode amplitudes $j_{lm}^{(1)}(t)$ and $j_{lm}^{(2)}(t)$, while its spatial dependence is described by vector SHs $\mathbf{\Psi}_{lm}(\hat{\mathbf{r}})$ and $\mathbf{\Phi}_{lm}(\hat{\mathbf{r}})$ (*Sandberg, 1978*) (Appendix 2, *Video 2*).

$$\mathbf{J}(\mathbf{r}, t) = \sum_{l=1}^{l_{\max}} \sum_{m=-l}^{l} \left( j_{lm}^{(1)}(t) \mathbf{\Psi}_{lm}(\hat{\mathbf{r}}) + j_{lm}^{(2)}(t) \mathbf{\Phi}_{lm}(\hat{\mathbf{r}}) \right). \tag{5}$$

Besides the in-plane divergence $\nabla_{\mathcal{S}} \cdot \mathbf{J}$ that leads to local density changes (see *Equation 1*), the cell number flux $\mathbf{J}(\mathbf{r}, t)$ also contains an in-plane curl component $\nabla_{\mathcal{S}} \times \mathbf{J}$ that is associated with locally rotational cell flux. The two sets of vector SHs $\{\mathbf{\Psi}_{lm}\}$ and $\{\mathbf{\Phi}_{lm}\}$ conveniently decompose the flux into these contributions: Because $\nabla_{\mathcal{S}} \cdot \mathbf{\Phi}_{lm} = \nabla_{\mathcal{S}} \times \mathbf{\Psi}_{lm} = 0$, and $\hat{\mathbf{r}} \cdot (\nabla_{\mathcal{S}} \times \mathbf{\Phi}_{lm}) = \nabla_{\mathcal{S}} \cdot \mathbf{\Psi}_{lm} = -l(l+1)Y_{lm}/R_s$ (*Sandberg, 1978*), we see from *Equation 5* that $j_{lm}^{(1)}(t)$ corresponds to modes that drive density changes and $j_{lm}^{(2)}(t)$ represents modes of local rotational cell motion that change relative cell positions but do not change local density. Indeed, using harmonic mode representations of the cell number density *Equation 4* and the cell number flux *Equation 5* directly in the continuity *Equation 1*, we find a set of ordinary differential equation in mode space

$$\frac{\mathrm{d}}{\mathrm{d}t} \rho_{lm}(t) = \frac{l(l+1)}{R_s} j_{lm}^{(1)}(t), \tag{6}$$

where $l = 0, 1, ..., l_{\max}$ and for each value of $l$, $m = -l, -l+1, ..., l-1, l$. *Equation 6* offers an alternative way of determining the modes $j_{lm}^{(1)}(t)$ directly from the modes $\rho_{lm}(t)$ of the coarse-grained cell number density (see *Equation 4* and *Equation 2a*), while ensuring that the resulting fields obey mass conservation exactly. In practice, the modes $j_{lm}^{(1)}(t)$ found from a vector harmonic representation of the coarse-grained cell number flux (*Equation 2b*) will often deviate from modes $j_{lm}^{(1)}(t)$ determined from *Equation 6*, even if cell numbers are expected to be conserved. This can be, for example, due to limited accuracy in determining velocities $\mathbf{v}_\alpha(t)$ from noisy single-cell trajectories $\mathbf{r}_\alpha(t)$, or due to spatially inhomogeneous appearances and disappearances of cells in tracking data. Consistent with our simplifying assumption that cell number changes in the data can be sufficiently well approximated by a globally homogeneous growth rate (compare *Equation 1* with *Equation 3*), the subsequent analysis uses the modes $j_{lm}^{(1)}(t)$ as determined from the density modes $\rho_{lm}(t)$ via *Equation 6*, together with modes $j_{lm}^{(2)}(t)$ from the explicit velocity coarse-graining *Equation 2b*. The complete construction is detailed in Appendix 2 and the full coarse-grained dynamics is shown in *Video 1*.

The representation of $\rho(\mathbf{r}, t)$ and $\mathbf{J}(\mathbf{r}, t)$ in terms of spherical harmonic modes with $l \leq l_{\max}$ leads in total to $3(l_{\max} + 1)^2$ mode amplitude trajectories, displaying only a few dominant contributions (*Figure 1D*) with almost no signal remaining for $l \geq 5$ (*Figure 1—figure supplement 1*, *Video 2*). This demonstrates that the underlying coarse-grained experimental data is sufficiently smooth and implies that a spectral representations is indeed meaningful. Thus, the coarse-graining approach outlined above provides a sparse spectral representation of high-dimensional microscopic single-cell data. The associated harmonic basis functions and vectors have an intuitive physical meaning, convenient algebraic properties and, as we will see, encode information about the length scales and symmetries of the collective dynamics.

## Temporal mode representation

We further compress the dynamical information by representing the time series of the modes in terms of Chebyshev polynomial basis functions $T_n(t)$ (*Driscoll et al., 2014*; *Mason and Handscomb, 2002*). To simplify notation, we define a dynamic mode vector $\mathbf{a}(t) = [\rho_{lm}(t), j_{lm}^{(1)}(t), j_{lm}^{(2)}(t)]^\top$ that collects all the modes up to $l = l_{\max}$ determined in the previous section and consider an expansion

$$\mathbf{a}(t) = \sum_{n=0}^{n_{\max}} T_n(t)\, \hat{\mathbf{a}}_n \tag{7}$$

in terms of the spatio-temporal mode coefficients $\hat{\mathbf{a}}_n$ with temporal mode number $n$ (Appendix 2). This compression allows us to accurately evaluate time derivatives of the mode amplitudes (*Supekar et al., 2021*), an important step when using *Equation 6* to determine flux modes $j_{lm}^{(1)}(t)$ directly from density modes $\rho_{lm}$. Fixing $l_{\max} = 4$ and $n_{\max} = 30$ in the remainder, the initial single-cell data set of about 1.4 million recorded cell position entries, or 4.2 million degrees of freedom, has thus been reduced to 2250 mode coefficients, corresponding to a compression ratio $\gtrsim 1800$.

## Characterization of the developmental mode dynamics

A harmonic mode decomposition naturally integrates the geometry of the underlying domain and simultaneously provides useful insights into spatial scales and symmetries of the dynamics. For each mode $(lm)$ in the sets of SHs $\{Y_{lm}\}$, $\{\mathbf{\Psi}_{lm}\}$ and $\{\mathbf{\Phi}_{lm}\}$, the integer index $l$ indicates the spatial scale of the harmonic, with $l = 0$ being a constant and larger $l$ indicating progressively finer spatial scales. The second index $m \in \{-l, -l+1, \ldots, l\}$ provides additional information about the orientation of the harmonic scalar function or vector field. The modes $l = 1$ and $l = 2$ are particularly useful for characterizing the symmetry of spatial patterns on a spherical surface (*Mietke et al., 2019*; *Scholich et al., 2020*): Modes with $l = 1$ indicate patterns with a global polar symmetry, whereas modes with $l = 2$ represent spatial patterns with a global nematic symmetry. We now exploit these features for a detailed characterization of the symmetry breaking that takes place during cellular rearrangements and to study the properties of the cellular flux in more detail. To this end, we discuss spatial averages

$$\langle O \rangle_s(t) = \tfrac{1}{A_s} \int_{\mathcal{S}} dA_s\, O(\mathbf{r}, t) \tag{8}$$

of different real-space observables $O(\mathbf{r}, t)$ over the mid-surface $\mathcal{S}$.

### Mode signatures of developmental symmetry breaking

To study how different developmental stages and their associated symmetry breaking events are reflected in the mode representation, we first consider the average cell surface density fluctuations

$$\left\langle \left( \rho - \langle \rho \rangle_s \right)^2 \right\rangle_s = \sum_{l=1}^{l_{\max}} \sum_{m=-l}^{l} \rho_{lm}^2(t). \tag{9}$$

For each mode $l$, the power spectrum $P_{\rho,l}(t) = \sum_{m=-l}^{l} \rho_{lm}^2(t)$ in *Equation 9* provides a rotationally invariant quantity (*Çetingül et al., 2012*; *Schwab et al., 2013*) that can effectively serve as an order parameter to characterize the symmetry of cell density patterns on the spherical surface. The dynamics of the density fluctuations given in *Equation 9* broken down into contributions $P_{\rho,l}(t)$ from each mode $l \leq l_{\max} = 4$ is shown in *Figure 2B*. Several features of this representation are particularly striking and can be directly related to specific developmental stages. First, patterns of cell surface density fluctuations evolve from a dominantly polar symmetry ($l = 1$) into density patterns with a prominent nematic symmetry ($l = 2$). These mode signatures intuitively reflect the essential symmetry breaking that takes place when cells collectively reorganize from an initially localized cell dome (*Figure 1B*, 52 min) into an elongated shape that wraps in an open ring-like pattern around the yolk cell (*Figure 1B*, 760 min). Second, during this transition at around 300 min (9 hpf) (black triangle in *Figure 2B*), the cell surface density is most homogeneous as fluctuations become minimal for all modes $l$. Interestingly, this time point approximately marks the completion of epiboly, when the different cell layers have fully engulfed the yolk. Finally, although in a less pronounced manner, the power spectrum of the mode $l = 4$ also exhibits an increased amplitude towards later times, indicating the formation of structures at finer spatial scales as development progresses. We find that mode signatures of the symmetry

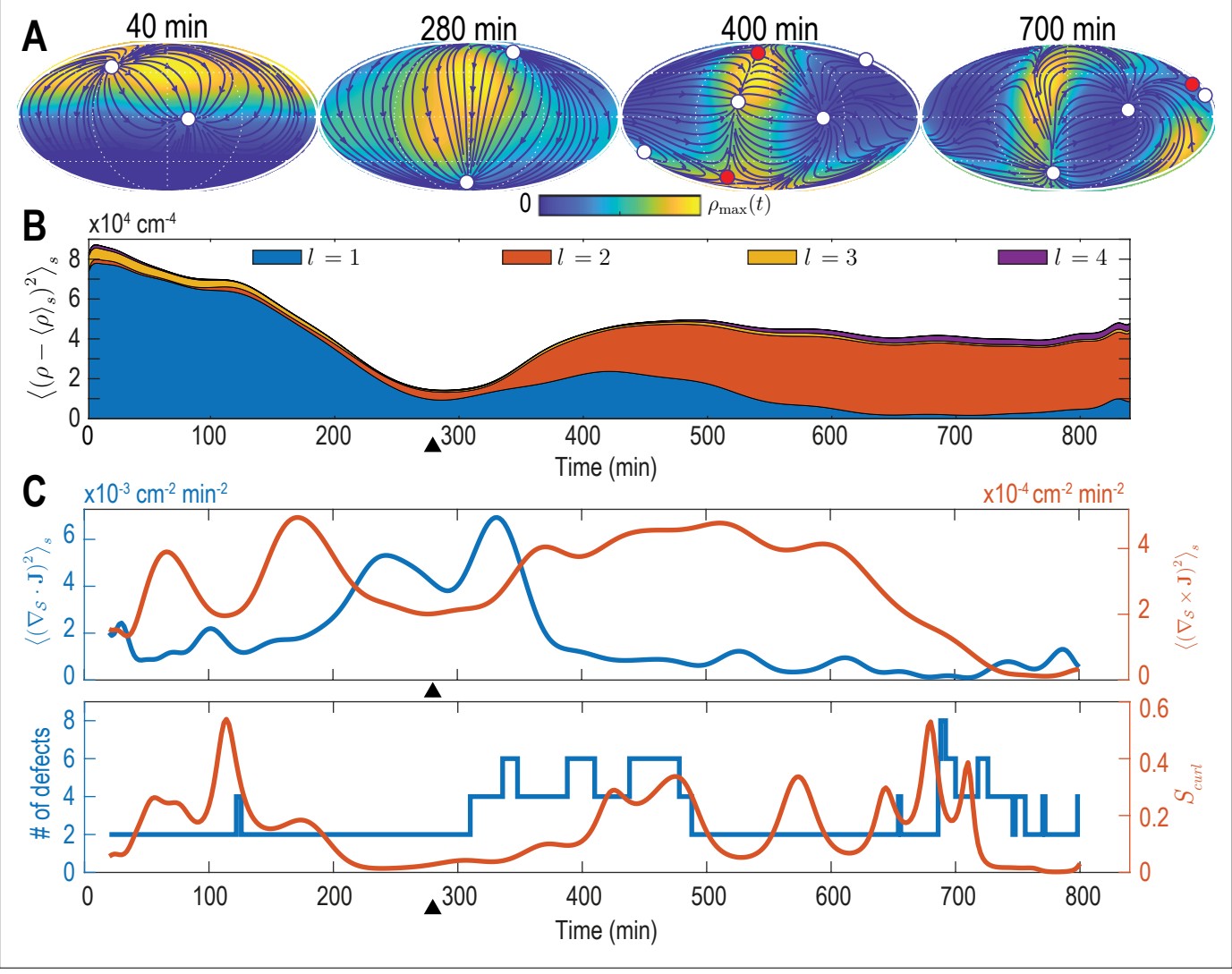

**Figure 2.** Mode signatures of developmental symmetry breaking and topological defects in cellular flux. (**A**) Two-dimensional Mollweide projection of the compressed coarse-grained density field $\rho(\mathbf{r}, t)$ (colormap) and of the coarse-grained cell flux $\mathbf{J}(\mathbf{r}, t)$ (streamlines) at different time points of zebrafish gastrulation. White circles depict topological defects of charge +1 in the flux vector field, red circles depict defects with charge -1. The total defect charge is 2 at all times. Defects are seen to 'lead' the large-scale motion of cells and later localize mostly along the curve defined by the forming spine. Animal pole (AP) and ventral pole (VP) are located at top and bottom, respectively. (**B**) Density fluctuations as a function of developmental time (see *Equation 9*), broken down in contributions from different harmonic modes $l$. The underlying symmetry breaking is highlighted prominently by this representation: During the first 75% of epiboly (0–280 min) cells migrate away from, but are still mostly located near the animal pole, presenting a density pattern with polar symmetry ($l = 1$). During the following convergent extension phase cells converge towards a confined elongated region that is 'wrapped' around the yolk, corresponding to a density pattern with nematic symmetry ($l = 2$). Black triangles indicate transition from epiboly to convergent extension. (**C**) Comparison of surface averaged divergence $\nabla_\mathcal{S} \cdot \mathbf{J}$ and curl $\nabla_\mathcal{S} \times \mathbf{J}$ of the cellular flux computed via *Equation 10a* and *Equation 10b*, respectively (top). A relative curl amplitude $S_{curl}$ computed from these quantities via *Equation 11* correlates with the appearance of an increased number of topological defects in the cell flux (bottom), suggesting that incompressible, rotational cell flux is associated with the formation of defects.

The online version of this article includes the following figure supplement(s) for figure 2:

**Figure supplement 1.** Analysis of the harmonic mode representation for a second experimental dataset.

**Figure supplement 2.** Validation of automated defect tracking.

**Figure supplement 3.** Analysis of fluxes and defects for different coarse-graining length scales (Sample 1).

**Figure supplement 4.** Analysis of fluxes and defects for different coarse-graining length scales (Sample 2).

breaking and progression through developmental stages are robust (*Figure 2—figure supplement 1B, D*), illustrating that mode-based analysis can provide a systematic and meaningful characterization of developmental symmetry breaking events.

## Mode signatures of emergent topological defects in cellular flux

The vectorial nature of the cell number flux $\mathbf{J}(\mathbf{r}, t)$ on a spherical surface implies the presence of topological defects (colored circles in *Figure 2A*, see Materials and methods) (*Kamien, 2002*). Several recent experimental results pertaining to the self-organization of multicellular systems suggest an important role of such topological defects in organizing morphogenetic events (*Doostmohammadi et al., 2016*; *Saw et al., 2017*; *Guillamat et al., 2020*; *Copenhagen et al., 2020*; *Meacock et al., 2020*; *Maroudas-Sacks et al., 2020*). We therefore analyze how defects within the cell number flux $\mathbf{J}(\mathbf{r}, t)$ are dynamically organized during early zebrafish gastrulation and if signatures of defect formation and annihilation are present in the mode representation *Equation 5*. We first consider the average squared divergence and curl of the cell number flux given by

$$\left\langle \left( \nabla_{\mathcal{S}} \cdot \mathbf{J} \right)^2 \right\rangle_s = \sum_{l=1}^{l_{\max}} \sum_{m=-l}^{m} \left[ \frac{l(l+1)}{R_s} j_{lm}^{(1)}(t) \right]^2, \tag{10a}$$

$$\left\langle \left( \nabla_{\mathcal{S}} \times \mathbf{J} \right)^2 \right\rangle_s = \sum_{l=1}^{l_{\max}} \sum_{m=-l}^{m} \left[ \frac{l(l+1)}{R_s} j_{lm}^{(2)}(t) \right]^2, \tag{10b}$$

which are shown in *Figure 2C* (top). The two contributions to the collective cellular dynamics – locally compressible, divergent flux quantified by the divergence $\nabla_{\mathcal{S}} \cdot \mathbf{J}$ and locally incompressible, rotational cell motion characterized by the curl $\nabla_{\mathcal{S}} \times \mathbf{J}$ – are independently determined by the modes $j_{lm}^{(1)}(t)$ and $j_{lm}^{(2)}(t)$. Therefore, each contribution can be evaluated conveniently and with high accuracy from a representation of $\mathbf{J}(\mathbf{r}, t)$ in terms of vector SHs. From *Figure 2C* (top), we see that the most significant divergent flux (blue curve) occurs around 300 min at the transition from epiboly towards the convergence and extension stage. A quantification of the incompressible rotational flux relative to the total cell number flux is shown in *Figure 2C* (bottom), where we plotted the relative curl amplitude

$$S_{\text{curl}}(t) = \frac{\sum_{l,m} \left[ j_{lm}^{(2)}(t) \right]^2}{\sum_{l,m} \left[ j_{l,m}^{(1)}(t) \right]^2 + \sum_{l,m} \left[ j_{l,m}^{(2)}(t) \right]^2}. \tag{11}$$

This measure suggests a correlation between incompressible rotational cell motion and the occurrence of topological defects (circles in *Figure 2A*) in the cell flux $\mathbf{J}(\mathbf{r}, t)$. The total number of topological defects present at any time point is depicted in *Figure 2C* (bottom, blue curve). Because the vector-valued flux is defined on a sphere, we observe that the total topological charge always sums to +2 (*Kamien, 2002*), while additional defect pairs with opposite charge (red +1 and white -1 circles in *Figure 2A*) can be created, resulting in total defect numbers greater than two (see *Figure 2C*, bottom). Interestingly, the relative curl amplitude $S_{\text{curl}}$ defined in *Equation 11* indicates that increased contributions from incompressible rotational flux are associated with the formation of topological defects in the cell number flux, a feature that is robustly identified by our framework (*Figure 2—figure supplement 1A, C*, *Figure 2—figure supplement 3*, *Figure 2—figure supplement 4*). The appearance of additional defects at the end of epiboly, when the developing embryo begins to extrude more significantly in the radial direction, suggests that topological defects in the 2D projected cellular flux fields could signal the start of the formation of more complex structures in three dimensions.

## Learning a linear hydrodynamic model of the developmental mode dynamics

The results in *Figure 2* confirm that a low-dimensional mode representation can capture essential characteristics of developmental symmetry breaking processes. The mode representation therefore provides a natural starting point for the inference of hydrodynamic models from coarse-grained cell-tracking data. For a given time-dependent mode vector $\mathbf{a}(t) = [\rho_{lm}(t), j_{lm}^{(1)}(t), j_{lm}^{(2)}(t)]^\top$ that contains all modes up to $l = l_{\max}$, the simplest hydrodynamic model corresponds to the linear dynamical equation

$$\frac{\mathrm{d}\mathbf{a}(t)}{\mathrm{d}t} = M \cdot \mathbf{a}(t), \tag{12}$$

where the *constant* coefficient matrix $M$ encodes the couplings between different modes. Intuitively, *Equation 12* aims to describe an experimentally observed density and flux dynamics in terms of a relaxation process, starting from inhomogeneous initial conditions represented by $\mathbf{a}(0)$. The mathematical learning problem is then to find a coefficient matrix $M$ such that the linear model *Equation 12* holds for the mode vector time series $\mathbf{a}(t)$ that was determined from the coarse-graining procedure described in the previous sections.

## Validation of the learning framework using active Brownian particle dynamics

Before applying the combined coarse-graining and inference framework to experimental data, we illustrate and validate the learning approach on synthetic data for which coarse-graining results and hydrodynamic mean-field equations are analytically tractable. To this end, we consider the stochastic dynamics of non-interacting active Brownian particles (ABPs) on the unit sphere of radius $R_0 = 1$ (*Sknepnek and Henkes, 2015*; *Fily et al., 2016*; *Castro-Villarreal and Sevilla, 2018*). Similar to a migrating cell, an ABP at position $\mathbf{x}(t)$ moves across the unit sphere at constant speed $v_0$ in the direction of its fluctuating orientation unit vector $\mathbf{u}(t)$. The strength of the orientational Gaussian white noise is characterized by a rotational diffusion constant $D_r$ (*Figure 3A*, Appendix 3).

Compared with conventional passive Brownian motion, self-propulsion of an ABP along its orientation direction $\mathbf{u}$ introduces a persistence to the particle's motion that is reduced as rotational noise $D_r$ is increased. Additionally, the topology of the spherical surface implies that in the low-noise regime, $R_0 D_r / v_0 < 1$, particles are expected to return to the vicinity of their starting points after a duration $\Delta t \approx 2\pi R_0 / v_0$. The conjunction of persistent motion and topology then leads to oscillatory dynamics in the positional correlation $\langle \mathbf{x}(t) \cdot \mathbf{x}(0) \rangle$ (blue dots in *Figure 3B*, Appendix 3). Comparing correlations from stochastic ABP simulations in different noise regimes with theoretical predictions (solid lines in *Figure 3B*) validates our numerical ABP simulation scheme.

To generate a test data set for our coarse-graining and inference framework, we simulated non-interacting ABPs in both the low-noise ($R_0 D_r / v_0 < 1$) and the high-noise ($R_0 D_r / v_0 > 1$) regime with initial positions drawn from the experimental data shown in *Figure 1*. Specifically, at each cell position present in the data, we generated 60 particles with random orientation, amounting to approximately $1.2 \times 10^5$ particles in total, and simulated their dynamics on a unit sphere. The resulting trajectory data were coarse-grained following the procedure outlined in the previous sections, yielding dynamic density fields $\rho(\mathbf{r}, t)$ and fluxes $\mathbf{J}(\mathbf{r}, t)$ (*Video 3*), together with their mode representations $\rho_{lm}(t), j_{lm}^{(1)}(t)$ and $j_{lm}^{(2)}(t)$.

In the second 'learning' step, we infer a sparse mode coupling matrix $M$ that approximates the dynamics *Equation 12* for the dynamical mode vectors $\mathbf{a}(t) = [\rho_{lm}, j_{lm}^{(1)}, j_{lm}^{(2)}]^\top$ obtained from the coarse-grained simulated ABP data. Our inference algorithm combines adjoint techniques (*Rackauckas et al., 2021*) and a multi-step sequential thresholding approach inspired by the Sparse Identification of Nonlinear Dynamics (SINDy) algorithm introduced by *Brunton et al., 2016*. The full algorithm is detailed in Appendix 4 and illustrated in the summary flowchart *Appendix 4—figure 1*. Importantly, we perform the sparse regression using dynamical mode vectors $\mathbf{a}(t)$ rescaled by their median absolute deviation (MAD) to compensate for substantial scale variations between different modes. The final output matrix $M$ of this learning algorithm is shown in the right panel of *Figure 3C* and can be compared against the analytically coarse-grained dynamics of ABPs on curved surfaces (*Fily et al., 2016*; *Castro-Villarreal and Sevilla, 2018*). Under suitable closure assumptions (Appendix 3), the mean-field dynamics of ABPs on a unit sphere is given in harmonic mode space by

$$\frac{\mathrm{d}\rho_{lm}}{\mathrm{d}t} = \frac{l(l+1)}{R_0} j_{lm}^{(1)} \tag{13a}$$

$$\frac{\mathrm{d}j_{lm}^{(1)}}{\mathrm{d}t} = -\frac{v_0^2}{2R_0} \rho_{lm} - D_r j_{lm}^{(1)} \tag{13b}$$

$$\frac{\mathrm{d}j_{lm}^{(2)}}{\mathrm{d}t} = -D_r j_{lm}^{(2)} \tag{13c}$$

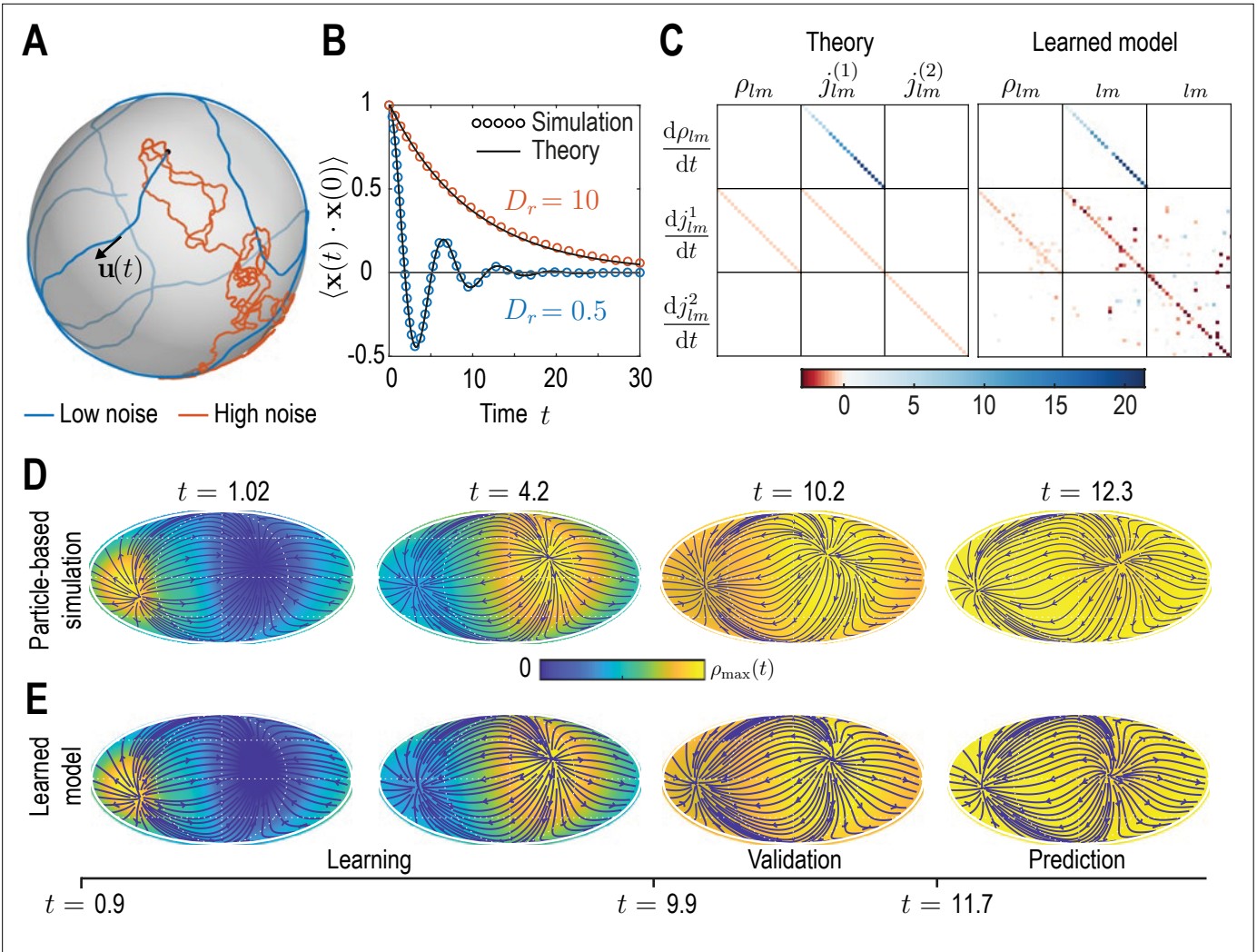

**Figure 3.** Learning active Brownian particle (ABP) dynamics on a sphere. (**A**) ABPs move on a unit sphere (radius $R_0 = 1$) with angular speed $v_0 = 1$ along a tangential unit vector $\mathbf{u}(t)$ that is subject to stochastic in-plane fluctuations (see Appendix 3 for further details). Example single-particle trajectories are shown in the high-noise (orange, $D_r = 10$ in units of $R_0 v_0$) and in the low-noise regime (blue, $D_r = 0.5$). Time $t$ is measured in units of $R_0/v_0$ in all panels. (**B**) Position correlation function $\langle \mathbf{x}(t) \cdot \mathbf{x}(0) \rangle$ averaged over $3 \times 10^4$ independent ABP trajectories show distinct oscillations of period $\approx 2\pi$ in the low-noise regime, as ABPs orbit the spherical surface more persistently (see *Video 3*). Standard error of the mean is smaller than symbol size. (**C**) Analytically predicted (left) and inferred (right) dynamical matrices $M$ (see *Equation 12*) describing the mean-field dynamics of a large collection of non-interacting ABPs (see *Equation 13a*, *Equation 13b*, *Equation 13c* and Appendix 3) show good quantitative agreement. (**D**) Mollweide projections of coarse-grained ABP simulations with $v_0 = 1$ and $D_r = 0.5$ using cell positions from the first time point in the zebrafish data (*Figure 1*) as the initial condition: At each position 60 particles with random orientation were generated and their ABP dynamics simulated, amounting to approximately $1.2 \times 10^5$ particles in total. The density fields homogenize over time, where the maximum density at $t = 12.3$ has decayed to about 5% of the maximum density at $t = 1.02$. Blue lines and arrows indicate streamlines of the cell flux $\mathbf{J}(\mathbf{r}, t)$. (**E**) Simulation of the learned linear model, *Equation 12* with $M$ shown in (C) (right), for the same initial condition as in D. Marked time points indicate intervals of learning, validation and prediction phases of the model inference (see Appendix 4).

from which we can read off the mode coupling matrix $M$ shown in the left panel of *Figure 3C*. A direct comparison between the theoretical and the inferred matrices shows that our framework recovers both the structure and the quantitative values of $M$ with good accuracy. Due to the finite number of ABPs used to determine the coarse-grained fields, we do not expect that the theoretically predicted coupling matrix is recovered perfectly from the data. Instead, some mode couplings suggested by *Equation 13a* may not be present or modified in the particular realization of the ABP dynamics that was coarse-grained. Indeed, direct simulation of the learned model projected in real space (*Figure 3E*) reveals a density and flux dynamics that agrees very well with the dynamics of the the coarse-grained

### Learning developmental mode dynamics from single-cell trajectories

Nicolas Romeo, Alasdair D. [...]xander Mietke, Jörn Dunkel

### Movie S3

**Video 3.** Coarse-grained dynamics of active Brownian particles on the unit sphere in the low-noise ($D_r = 0.5$) and high-noise ($D_r = 10$) regime. Data from $N = 3 \times 10^4$ independent ABP simulations was coarse-grained using the kernels $f_k(\omega)$ and $g_k(\omega)$ ($k = 6$) described in Appendix 1. Initial ABP positions were sampled from an axisymmetric distribution with $p(\theta) \propto \cos\theta\, \mathbf{1}_{\{\theta < \pi/2\}}$. Mollweide projections in the left and right column are color-coded for density and flux magnitude $|\mathbf{J}(\mathbf{r}, t)|$, respectively. Colormaps are normalized by the maximum values of density and flux magnitude fields across all time points.

https://elifesciences.org/articles/68679/figures#video3

input data (*Figure 3D*). Altogether, these results demonstrate that the proposed inference framework enables us to to faithfully recover expected mean-field dynamics from coarse-grained fields of noisy particle-based data.

## Learning developmental mode dynamics from experimental data

The same inference framework can now be directly applied to the coarse-grained experimental zebrafish embryo data shown in *Figure 1C and D*, yielding a sparse coefficient matrix $M$ (*Figure 4A and B*) that encodes the dynamics of the developmental mode vector $\mathbf{a}(t) = [\rho_{lm}(t), j_{lm}^{(1)}(t), j_{lm}^{(2)}(t)]^\top$ according to *Equation 12*. The inferred coupling between the time derivative of density modes $\rho_{lm}$ and flux modes $j_{lm}^{(1)}$ faithfully recovers mass conservation (*Figure 4C*; see *Equation 6*). Overall, the learned matrix $M$ has 395 non-zero elements, effectively providing further compression of the experimental data, which required 2,250 spatio-temporal mode coefficients collected in $\hat{\mathbf{a}}_n$ (see *Equation 7*) for its representation. Using the mode vector $\mathbf{a}(t = 0)$ of the first experimental time point as the initial condition, the inferred minimal model *Equation 12* with $M$ shown in (*Figure 4A and B*) faithfully recovers both the mode and real-space dynamics seen in the coarse-grained fields of the experimental input data (*Figure 4E–G*, *Video 4*).

It is instructive to analyze the inferred matrix $M$ and the linear model it encodes in more detail. Comparing the MAD-rescaled matrix (see Appendix 4) learned for the experimental zebrafish data (*Figure 4B*) with the non-dimensionalized matrix learned for the active Brownian particle dynamics (*Figure 3C*), we find similar patterns of prominent diagonal and block-diagonal couplings. Consistent with the analysis of single cell trajectories (*Shah et al., 2019*), this suggests that a random, but persistent movement of cells akin to ABPs moving on a sphere partially contributes to the early gastrulation process in zebrafish. This is complemented in the minimal model of the experimental dynamics by significant off-diagonal contributions (*Figure 4B*), which are absent in the non-interacting ABP model. Such off-diagonal contributions represent effective linear approximations of cell-cell interactions, environmental influences or other external stimuli reflected in the experimental time-series data. Ultimately, such contributions to the mode coupling matrix $M$ help realize the symmetry breaking process observed in the underlying experimental data (*Figure 2*).

The inferred mode coupling matrix $M$ shown in *Figure 4B* together with *Equation 12* provides a highly robust minimal model. Specifically, despite being linear, it is numerically stable over a period approximately four times as long as the input data from which the matrix $M$ was learned. Furthermore, simulations with modified initial conditions (see *Figure 4—figure supplement 1*) still exhibit a characteristic symmetry breaking and lead to the emergence of density and flux patterns similar to those seen in *Figure 4F and G*. For example, simulating *Equation 12* using the initial condition of a different experimental data set (*Figure 2—figure supplement 1*) leads to final patterns with the same symmetry as in the original training data, further corroborating that the observed symmetry breaking is directly encoded in the interactions represented by the matrix $M$. A similar robustness is observed under moderate perturbations of the initial condition, such as a rotation of initial cell density patterns relative to the coordinate system in which $M$ was inferred, or a local depletion of the initial density, emulating a partial removal of cells as experimentally realized in *Morita et al., 2017*. Taken together, these numerical experiments demonstrate that the inferred mode coupling matrix $M$ meaningfully captures the dynamics and interactions of cells that facilitate the symmetry breaking observed during early zebrafish development.

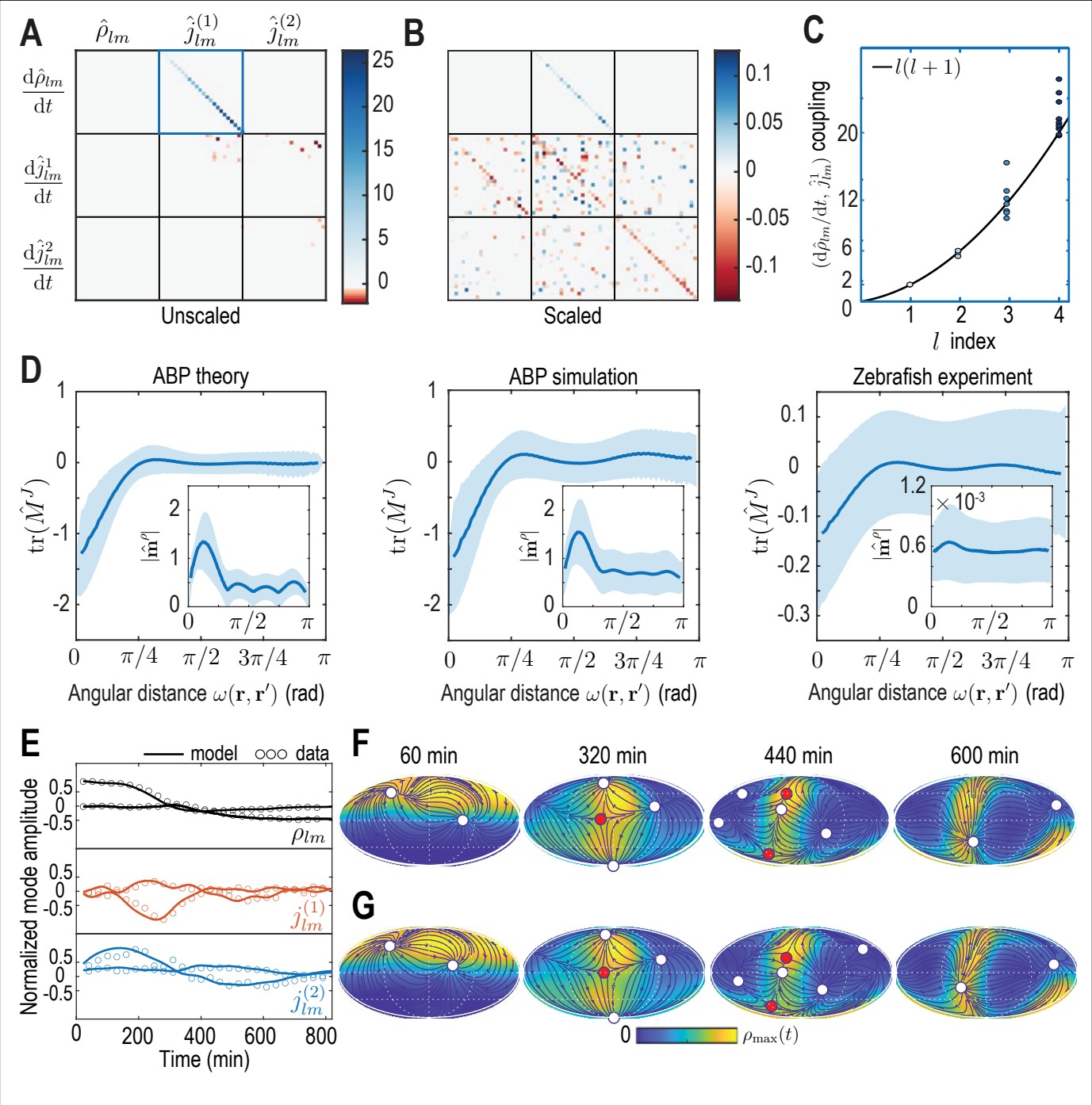

**Figure 4.** Model learning for experimental data of collective cell motion during early zebrafish development. (**A**) Visualization of the constant mode coupling matrix $M$ that was learned from experimental data (see Appendix 4) and describes the dynamics of the mode vector $\mathbf{a} = [\rho_{lm}(t), j_{lm}^{(1)}(t), j_{lm}^{(2)}(t)]^T$ via *Equation 12*. Dimensionless fields are defined by $\hat{\rho}_{lm} = R_s^2 \rho_{lm}$ and $\hat{j}_{lm}^{(i)} = R_s \Delta t j_{lm}^{(i)}$ ($i = 1, 2$) with $R_s = 300\,\mu$m and $\Delta t = 2$ min. (**B**) Scaling the learned matrix $M$ by the Mean Absolute Deviation (MAD) of the modes (see Appendix 4) reveals structures reminiscent of the mode coupling matrix learned for ABPs (*Figure 3C*). (**C**) The learned model recovers mass conservation in mode space (*Equation 6*). (**D**) Comparison of theoretical and inferred real-space kernels (see *Equation 14* and Appendix 4) for the ABP dynamics and for the experimental data of collective cell motion. The trace of the non-dimensional kernel $\hat{M}^J(\mathbf{r}, \mathbf{r}')$ (the only non-zero eigenvalue, *Appendix 4—figure 2*) indicates a localized flux-flux coupling with a similar profile among both systems. The oscillating magnitude of the non-dimensionalized density-flux kernel $|\hat{\mathbf{m}}^\rho(\mathbf{r}, \mathbf{r}')|$ (insets) in the ABP system indicates a gradient-like coupling and is consequence of the persistent ABP motion. In the experimental data, a first peak around $\omega = \pi/4$ is also visible, but less pronounced. All kernel properties were computed by averaging over pairs of positions $\mathbf{r}, \mathbf{r}'$ that are separated by the same angular distance $\omega = \arccos(\mathbf{r} \cdot \mathbf{r}') \in [0, \pi]$. Solid lines indicate mean, shaded areas indicate standard deviation. (**E**) Comparison of experimental

*Figure 4 continued on next page*

*Figure 4 continued*

mode dynamics (circles) with numerical solution (solid line) of the minimal model *Equation 12* for learned matrix $M$ visualized in A. For clarity, the comparison is shown for the two dominant modes of each set of harmonic modes $\rho_{lm}$, $j_{lm}^{(1)}$ and $j_{lm}^{(2)}$. (F, G) Mollweide projections of the experimental data (F) and of the numerical solution of the learned model (G) show very good agreement (*Video 4*). Blue lines and arrows illustrate streamlines defined by the cell flux $\mathbf{J}(\mathbf{r}, t)$, circles depict defects with topological charge +1 (white) and -1 (red).

The online version of this article includes the following figure supplement(s) for figure 4:

**Figure supplement 1.** Simulating the learned model with different initial conditions.

## Green's function representation of learned models in real space

To characterize the inferred spatial interactions in more detail, we can analyze the real-space representation of the learned mode coupling matrix $M$. While the density dynamics represented by $M$ (the first row in *Figure 4AB*) simply reflects mass conservation *Equation 1* in real space, the dynamics of the flux (the second and third row in *Figure 4A and B*) corresponds in real space to the integral equation (Appendix 4)

$$\frac{\partial}{\partial t}\mathbf{J}(\mathbf{r}, t) = \int d\Omega' \left[ \mathbf{m}^\rho(\mathbf{r}, \mathbf{r}')\rho(\mathbf{r}', t) + M^J(\mathbf{r}, \mathbf{r}') \cdot \mathbf{J}(\mathbf{r}', t) \right],$$  (14)

where $d\Omega' = \sin\theta' d\theta' d\phi'$ is the spherical surface area element. The vector-valued kernel $\mathbf{m}^\rho(\mathbf{r}, \mathbf{r}')$ in *Equation 14* connects the distribution of cell density $\rho$ across the surface to dynamic changes of the flux $\mathbf{J}$ at a given point $\mathbf{r}$. Similarly, the matrix-valued kernel $M^J(\mathbf{r}, \mathbf{r}')$ describes how the distribution of cell fluxes at $\mathbf{r}'$ affects temporal changes of the flux at $\mathbf{r}$.

To analyze the spatial range of interactions between points $\mathbf{r}$ and $\mathbf{r}'$, we use the fact that the matrix-valued kernel $M^J(\mathbf{r}, \mathbf{r}')$ has only one non-zero eigenvalue (*Appendix 4—figure 2*). Consequently, the trace $\text{tr}(M^J)$ serves as a proxy for the distance-dependent interaction strength mediated by $M^J$. Averages of $\text{tr}(M^J)$ over point-pairs with the same angular distance $\omega = \text{acos}(\mathbf{r} \cdot \mathbf{r}')$ are shown for the ABP dynamics and for the minimal model inferred from experimental data in *Figure 4D*. Note that to make the models amenable to comparison, we compute $M^J(\mathbf{r}, \mathbf{r}')$ from the known mean-field model of ABPs (*Equation 13a*) using the same *finite* number of modes as used to represent the ABP and the zebrafish data ($l_{\max} = 4$). In theory, one expects for the ABP dynamics a highly localized, homogeneous kernel $\text{tr}(M^J) \sim \delta(\mathbf{r} - \mathbf{r}')$, so that an exact spectral representation would require an infinite number of modes (see Appendix 4). In practice, using a finite number of modes leads to a wider kernel range (*Figure 4D* 'ABP theory') and introduces an apparent spatial inhomogeneity, as indicated by the non-zero standard deviation of $\text{tr}(M^J)$ at fixed distance $\omega$ (blue shades). Both the quantitative profile of $\text{tr}(M^J)$ and its variation are successfully recovered by applying the inference framework to stochastic simulations of ABPs (*Figure 4D* 'ABP simulation') where $M^J(\mathbf{r}, \mathbf{r}')$ was computed from the learned mode coupling matrix $M$ shown in *Figure 3C*. For the inferred minimal model of the cell dynamics (*Figure 4D* 'Zebrafish experiment'), we find a similar short-ranged flux-flux coupling mediated by $M^J$. However, the increased variability of $\text{tr}(M^J)$ at fixed distances $\omega$ indicates more substantial spatial inhomogeneities of the corresponding interactions. These inhomogeneities are absent in a non-interacting system of ABPs and represent an interpretable real-space signature of the symmetry-breaking mechanisms built into the underlying mode coupling matrix $M$.

A similar analysis can be performed for the kernel $\mathbf{m}^\rho(\mathbf{r}, \mathbf{r}')$ that couples the density at position $\mathbf{r}'$ to dynamics of fluxes at position $\mathbf{r}$ (see

**Learning developmental mode dynamics from single-cell trajectories**

Nicolas Romeo, Alasdair D. ...xander Mietke, Jörn Dunkel

**Movie S4**

**Video 4.** Comparison of dynamics of the experimental and learned density $\rho(\mathbf{r}, t)$ (colormap) and flux fields $\mathbf{J}(\mathbf{r}, t)$ (streamlines) represented in a Mollweide projection. White circles depict topological defects of charge +1 in the vector field $\mathbf{J}(\mathbf{r}, t)$, red circles depict defects with charge -1. The total defect charge is 2 at all times. Top row depicts the coarse-grained (see *Equation 2a*) and projected (see *Equations 4–7*) experimental data, snapshots in the bottom row are obtained by reintegrating the ordinary differential equation model *Equation 12* using the learned matrix $M$ (see Figure 4A). The colorbar is at each time point scaled to the interval $[0, \max_{\mathbf{r}}\rho(\mathbf{r}, t)]$.

https://elifesciences.org/articles/68679/figures#video4

*Equation 14*), where we average the magnitude $|\mathbf{m}^\rho(\mathbf{r}, \mathbf{r}')|$ over pairs $(\mathbf{r}, \mathbf{r}')$ with the same angular distance $\omega$ (*Figure 4D* insets). Using a finite number of modes to compute this kernel in the different scenarios again introduces apparent spatial inhomogeneities in all cases. Additionally, all kernel profiles exhibit a distinct maximum at short range, indicating a coupling between density gradients and the flux dynamics that emerges microscopically from a persistent ABP and cell motion (see Appendix 3 and 4) – an observations that is consistent with the similar block-diagonal structure of both inferred matrices $M$ (compare *Figures 3C and 4B*).

In conclusion, the real-space analysis and comparison of inferred interaction kernels further highlights potential ABP-like contributions to the collective cellular organization during early zebrafish development and reveals an effectively non-local coupling between density and flux dynamics. The latter could result, for example, from unresolved fast-evolving morphogens (*Hannezo and Heisenberg, 2019*), through mechanical interactions with the surrounding material (*Münster et al., 2019*) or due to other relevant degrees of freedom that are not explicitly captured in this linear hydrodynamic model. More generally, a real-space representation of kernels provides an alternative interpretable way to study the interactions and symmetry-breaking mechanisms encoded by models directly learned in mode space.

## Discussion

Leveraging a sparse mode representation of collective cellular dynamics on a curved surface, we have presented a learning framework that translates single-cell trajectories into quantitative hydrodynamic models. This work complements traditional approaches to find quantitative continuum models of complex multicellular processes (*Etournay et al., 2015*; *Hannezo et al., 2015*; *Morita et al., 2017*; *Streichan et al., 2018*; *Münster et al., 2019*) that match problem-specific constitutive relations of active materials in real-space with experimental observations. We have demonstrated here that length scales and symmetries associated with a mode representation can directly inform about the character of symmetry breaking transitions and topological features of collective cellular motion even before a model is specified. The successful applications to synthetic ABP simulation data and experimental zebrafish embryo data show that model learning in mode space provides a promising and computationally feasible approach to infer quantitative interpretable models in complex geometries.

The learned linear minimal model for cell migration during early zebrafish morphogenesis quantitatively recapitulates the spatiotemporal dynamics of a complex developmental process (*Figure 4F and G*), and highlights similarities between collective cell migration and analytically tractable ABP dynamics on a curved surface. An extension to nonlinear mode-coupling models or an integration of additional, experimentally measured degrees of freedom, such as concentration fields of morphogens involved in mechanochemical feedbacks (*Hannezo and Heisenberg, 2019*), is in principle straightforward by including nonlinear terms in *Equation 12*. Furthermore, the above framework could be generalized to describe the dynamics within a spherical shell of finite height by complementing the surface vector SHs used in this work by their radial counterpart (*Barrera et al., 1985*).

To provide a concrete example, we focused here on applying the model learning framework to single-cell tracking data of early zebrafish morphogenesis. However, the essentially spherical organization of cells during gastrulation observed in zebrafish is shared by many species whose early development occurs through a similar discoidal cleavage (*Gilbert and Barresi, 2016*), and the framework introduced here is directly applicable once tracking data becomes available for these systems. More generally, as novel imaging technologies are being developed (*Keller et al., 2010*; *Royer et al., 2016*; *Shah et al., 2019*), we expect that even larger and more detailed imaging data will further facilitate the exploration of finer scales and length-scale bridging processes (*Lenne et al., 2021*) through learning approaches that directly built on mode-based data representations.

## Materials and methods
### Data pre-processing
We obtained two single-cell tracking data sets from the experiments described in *Shah et al., 2019*. These data consist of the Cartesian coordinates of each cell together with a tracking ID. Some of the data is accessible at https://idr.openmicroscopy.org with ID number idr0068. We first denoised each

cell trajectory using MATLAB's (*Matlab, 2019*) wavelet denoiser function wdenoise, and centered the cloud of cells by least-squares fitting a spherical surface through it and shifting the origin at each time to coincide with the center of this sphere. We then computed the velocity of each cell by using Tikhonov-regularized differentiation as described in *Knowles and Renka, 2014* and implemented in the MATLAB third-party module rdiff (*Wagner, 2020*). After examination of the cells' velocity distribution, we further removed outlier cells whose speed is in the 95th percentile or above and verified that this operation only removes aberrant cells. Finally, we rotated the data to align the animal pole of the embryo with the $z$-axis, as determined by the direction of the center of mass of the initial cell distribution. The resulting single cell data are shown as point clouds in *Figure 1B* and in *Video 1*.

### Topological defect tracking

We have developed a defect tracker that identifies topological defects in vector fields tangent to a spherical surface via integration along suitable Burger circuits. The corresponding software is available at (https://github.com/NicoRomeo/surf-vec-defects; *Romeo, 2022*, copy archived at swh:1:rev:6dc742c376b0d085e19ece65f932ac6935342aba).

## Acknowledgements

We thank Nico Scherf and Ghopi Shah for providing single-cell tracking data and for giving helpful advice on zebrafish development and we thank Paul Matsudaira for discussions. We thank the MIT SuperCloud (*Reuther et al., 2018*) for providing us access to HPC resources. This work was supported by a MathWorks Science Fellowship (NR and ADH), a Longterm Fellowship from the European Molecular Biology Organization (ALTF 528–2019, AM), a Postdoctoral Research Fellowship from the Deutsche Forschungsgemeinschaft (Project 431144836, AM), a Complex Systems Scholar Award from the James S McDonnell Foundation (JD), the Robert E Collins Distinguished Scholarship Fund (JD) and the Alfred P Sloan Foundation (G-2021–16758, JD).

## Additional information

### Funding

| Funder | Grant reference number | Author |
| --- | --- | --- |
| European Molecular Biology Organization | ALTF 528-2019 | Alexander Mietke |
| Deutsche Forschungsgemeinschaft | 431144836 | Alexander Mietke |
| James S. McDonnell Foundation | Complex Systems Scholar Award | Jörn Dunkel |
| Alfred P. Sloan Foundation | G-2021-16758 | Jörn Dunkel |
| MathWorks | | Nicolas Romeo Alasdair Hastewell |
| Robert E Collins Distinguished Scholarship Fund | | Jörn Dunkel |

The funders had no role in study design, data collection and interpretation, or the decision to submit the work for publication.

### Author contributions

Nicolas Romeo, Data curation, Formal analysis, Investigation, Methodology, Software, Validation, Writing – original draft, Writing – review and editing; Alasdair Hastewell, Methodology, Software, Writing – original draft, Writing – review and editing; Alexander Mietke, Conceptualization, Formal analysis, Methodology, Project administration, Supervision, Validation, Writing – original draft, Writing – review and editing; Jörn Dunkel, Conceptualization, Methodology, Project administration, Supervision, Writing – original draft, Writing – review and editing

## Author ORCIDs
Nicolas Romeo (ID) http://orcid.org/0000-0001-6926-5371
Alasdair Hastewell (ID) http://orcid.org/0000-0003-2633-380X
Alexander Mietke (ID) http://orcid.org/0000-0003-1170-2406
Jörn Dunkel (ID) http://orcid.org/0000-0001-8865-2369

## Decision letter and Author response
Decision letter https://doi.org/10.7554/eLife.68679.sa1
Author response https://doi.org/10.7554/eLife.68679.sa2

## Additional files

### Supplementary files
• Transparent reporting form

### Data availability
Raw data used in this study can be obtained at https://doi.org/10.1038/s41467-019-13625-0 and https://imb-dev.gitlab.io/cell-flow-navigator/.

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

# Appendix 1

## Consistent coarse-graining on curved surfaces

We describe the derivation of self-consistent coarse-graining kernels that are used in the main text to convert single cell information into a continuous density field and its associated fluxes on a spherical surface. We first motivate this problem for a flat surface and then proceed with a detailed derivation for the case of a spherical surface.

### Kernel consistency in Euclidean space

It is instructive to first consider a set of particles $\alpha = 1, 2, 3, \ldots$ at positions $\mathbf{X}_\alpha(t)$ moving with velocities $\mathbf{V}_\alpha(t) = \mathrm{d}\mathbf{X}_\alpha/\mathrm{d}t$, where capitalized vectors indicate position and velocity in *Euclidean* space, e.g. particles move on a flat surface or within some three-dimensional volume. A coarse-grained density $\rho(\mathbf{X}, t)$ and a mass flux $\mathbf{J}(\mathbf{X}, t)$ can be defined from this microscopic information by

$$\rho(\mathbf{X}, t) \quad = \sum_\alpha K_e\left[\mathbf{X}, \mathbf{X}_\alpha(t)\right], \tag{15a}$$

$$\mathbf{J}(\mathbf{X}, t) \quad = \sum_\alpha \mathcal{K}_e\left[\mathbf{X}, \mathbf{X}_\alpha(t)\right] \cdot \mathbf{V}_\alpha(t), \tag{15b}$$

where $K_e\left(\mathbf{X}, \mathbf{X}'\right)$ and $\mathcal{K}_e\left(\mathbf{X}, \mathbf{X}'\right)$ represent a scalar-valued and a matrix-valued kernel function, respectively. At the same time, in a system with a constant number of particles, mass conservation implies, in general,

$$\partial_t \rho(\mathbf{X}, t) + \nabla_\mathbf{X} \cdot \mathbf{J}(\mathbf{X}, t) = 0, \tag{16}$$

relating the density $\rho(\mathbf{X}, t)$ and the mass flux $\mathbf{J}(\mathbf{X}, t)$ of particles. Using the coarse-graining prescriptions *Equation 15a* and *Equation 15b* directly in *Equation 16* and assuming the resulting relation must hold for any set of particle trajectories, one finds a general kernel consistency relation

$$\nabla_{\mathbf{X}'} K_e(\mathbf{X}, \mathbf{X}') + \nabla_\mathbf{X} \cdot \mathcal{K}_e(\mathbf{X}, \mathbf{X}') = 0. \tag{17}$$

This condition is automatically satisfied for any translationally invariant and isotropic pair of kernels $K_e(\mathbf{X}, \mathbf{X}') = K_e(\mathbf{X} - \mathbf{X}')$ and $\mathcal{K}_e(\mathbf{X}, \mathbf{X}') = K_e(\mathbf{X} - \mathbf{X}')\mathbb{I}$, where  is the unit matrix. Coarse-graining with such kernels is frequently employed in practice: Positions and velocities can be, for example, simply convolved with a Gaussian function of mean zero (*Supekar et al., 2021*).

### Kernel consistency on a curved surface

For a surface parameterized by $\mathbf{r}(s^1, s^2) \in \mathbb{R}^3$ with generalized coordinates $s^1, s^2$, two tangential basis vectors are defined by $\mathbf{e}_i = \partial \mathbf{r}/\partial s^i$ $(i = 1, 2)$. Partial derivatives are, in the following, denoted $\partial_i := \partial/\partial s^i$. The metric tensor is given by $g_{ij} = \mathbf{e}_i \cdot \mathbf{e}_j$. The mean curvature is defined by $H\mathbf{n} = -\nabla_i \mathbf{e}^i/2$, where $\mathbf{n} = \mathbf{e}_1 \times \mathbf{e}_2/|\mathbf{e}_1 \times \mathbf{e}_2|$ denotes the unit surface normal and the Einstein summation convention is used. The covariant form of mass conservation *Equation 1* (main text) on a curved surface reads

$$\partial_t \rho + \nabla_i J^i = 0, \tag{18}$$

with $J^i = \mathbf{e}^i \cdot \mathbf{J}$ and $\nabla_i$ denotes the covariant derivative. In general, we are interested in describing an effective dynamics for cell positions and velocities that are projected onto a common reference sphere of radius $R_s$. Such a description can be found by first formulating the coarse-graining approach for a unit sphere, on which particle positions and velocities are fully determined by angular coordinates and corresponding angular velocities, and finally rescaling the density and flux fields by suitable factors of $R_s$. The corresponding coarse-graining *Equation 2b* (main text) of in-plane angular velocities $\bar{\mathbf{v}}_\alpha(t) = \mathbf{v}_\alpha(t)/|\mathbf{r}_\alpha(t)|$ for particles $\alpha$ on a unit sphere reads covariantly

$$J^i = \sum_\alpha \mathcal{K}\left(\mathbf{r}, \mathbf{r}_\alpha\right)^i_{j'} \bar{v}^{j'}_\alpha, \tag{19}$$

where $\bar{v}^i_\alpha = \mathbf{e}^i \cdot \bar{\mathbf{v}}_\alpha$ and we drop the dependence on time to simplify the notation. The two-point kernel tensor $\mathcal{K}\left(\mathbf{r}, \mathbf{r}'\right)_{ij'}$ (a 'bitensor') is evaluated in the tangent space of $\mathbf{r}$ for its first index and in the tangent space of $\mathbf{r}'$ at the second, primed index (*Appendix 1—figure 1*). Mass conservation on a curved surface, *Equation 18*, together with the coarse-graining prescriptions *Equation 2a* (main text) and *Equation 19* then implies a covariant kernel consistency relation

$$\partial_{j'} K(\mathbf{r}, \mathbf{r}') + \nabla_i \mathcal{K}(\mathbf{r}, \mathbf{r}')^i{}_{j'} = 0. \tag{20}$$

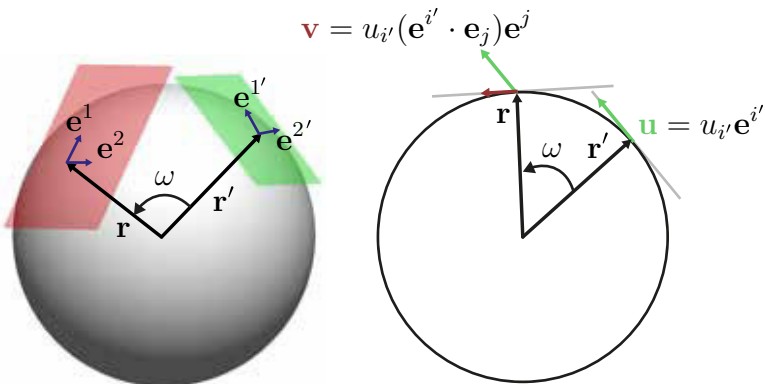

**Appendix 1—figure 1.** Illustration of the action of the coarse-graining tensor kernel $\mathcal{K}(\mathbf{r}, \mathbf{r}')_{ij'}$ *Equation 19*. Left: $\mathcal{K}_{ij'}$ acts in the two tangent space at points $\mathbf{r}$ and $\mathbf{r}'$ that are separated by an angular distance $\omega = \mathrm{acos}(\mathbf{r} \cdot \mathbf{r}')$. Each tangent plane has corresponding basis vectors, $\mathbf{e}_i$ $\mathbf{e}_{i'}$ for $i = 1, 2$. Right: The tensor kernel $\mathcal{K}_{ij'} \sim \mathbf{e}_i \cdot \mathbf{e}_{j'}$ projects vectors $\mathbf{u}$ in the tangent space of $\mathbf{r}'$ and generates a vector $\mathbf{v}$ tangent at $\mathbf{r}$.

## Solving the kernel consistency relation on a sphere

We solve *Equation 20* in the following on the unit sphere, such that $\mathbf{r} = \mathbf{n}$ corresponds to the surface normal. The final result can simply be rescaled to any spherical surface of radius $R_s$. Furthermore, we note that the parameter

$$x = \mathbf{r} \cdot \mathbf{r}' \tag{21}$$

provides a measure for the great circle distance $\omega(x) = \mathrm{acos}(x)$ between two points on a sphere. Hence, we consider an ansatz for the kernels in *Equation 20* of the form

$$K(\mathbf{r}, \mathbf{r}') = f(x) \tag{22a}$$

$$\mathcal{K}(\mathbf{r}, \mathbf{r}')_{ij'} = g(x)\mathbf{e}_i \cdot \mathbf{e}_{j'}, \tag{22b}$$

with two unknown scalar functions $f(x)$ and $g(x)$. The relevant derivatives of the ansatz *Equation 22a* and *Equation 22b* can readily be evaluated to

$$\partial_{j'} K(\mathbf{r}, \mathbf{r}') = \frac{\mathrm{d}f(x)}{\mathrm{d}x}\mathbf{r} \cdot \mathbf{e}_{j'} \tag{23a}$$

$$\nabla_i \mathcal{K}(\mathbf{r}, \mathbf{r}')^i{}_{j'} = \frac{\mathrm{d}g(x)}{\mathrm{d}x}\mathbf{r}' \cdot \left(\mathbf{e}_i \otimes \mathbf{e}^i\right) \cdot \mathbf{e}_{j'} - 2g(x)\mathbf{r} \cdot \mathbf{e}_{j'}. \tag{23b}$$

Here, $\otimes$ denotes a dyadic product and we use $\partial_i x = \mathbf{r}' \cdot \mathbf{e}_i$ and $\partial_{i'} x = \mathbf{r} \cdot \mathbf{e}_{i'}$, which follows from *Equation 21*, as well as $\nabla_i \mathbf{e}^i = -2\mathbf{r}$ in the second equation, which holds on a unit sphere and follows from the definition of the mean curvature. We then use the expansion of the identity matrix in $\mathbb{R}^3$ on the spherical basis $\mathbb{I} = \mathbf{e}_i \otimes \mathbf{e}^i + \mathbf{n} \otimes \mathbf{n}$, such that in our case with $\mathbf{r} = \mathbf{n}$ we have $\mathbf{e}_i \otimes \mathbf{e}^i = \mathbb{I} - \mathbf{r} \otimes \mathbf{r}$. Hence, *Equation 23b* becomes

$$\nabla_i \mathcal{K}(\mathbf{r}, \mathbf{r}')^i{}_{j'} = -\frac{\mathrm{d}g(x)}{\mathrm{d}x}(\mathbf{r}' \cdot \mathbf{r})(\mathbf{r} \cdot \mathbf{e}_{j'}) - 2g(x)\mathbf{r} \cdot \mathbf{e}_{j'}. \tag{24}$$

Using *Equation 24* and *Equation 23a* in the kernel consistency relation *Equation 20* and dividing by $\mathbf{r} \cdot \mathbf{e}_{j'}$ (at $\mathbf{r} = \mathbf{r}'$, for which $\mathbf{r} \cdot \mathbf{e}_{j'} = 0$, *Equation 20* is obeyed for any $f(x)$, $g(x)$), we find that the scalar functions in the kernel ansatz *Equation 22a* and *Equation 22b* have to obey

$$x\frac{\mathrm{d}g(x)}{\mathrm{d}x} + 2g(x) = \frac{\mathrm{d}f(x)}{\mathrm{d}x}.$$

Hence, the general covariant consistency relation *Equation 20* implies for the kernel ansatz *Equation 22a* and *Equation 22b* that the weighting functions $g(x)$ and $f(x)$ must be related by

$$g(x) = \frac{1}{x^2} \int_0^x \mathrm{d}u \, u \frac{\mathrm{d}f(u)}{\mathrm{d}u}. \tag{25}$$

## Kernel functions with compact support

In the last step, we determine a family of kernel functions $g(x)$ and $f(x)$ defined on the interval $x \in [-1, 1]$ that satisfy *Equation 25*, along with the requirements:

1. $f(x)$ and $g(x)$ must be $C^1$ regular on $[-1, 1]$
2. $f \geq 0$ on $[-1, 1]$
3. $f$ is normalized to one on the unit sphere.

Recalling $x = \cos[\omega(\mathbf{r}, \mathbf{r}')]$ with angular distance $\omega$ between $\mathbf{r}$ and $\mathbf{r}'$, a family of functions fulfilling these conditions is given by

$$f_k(\omega) = \frac{k+1}{2\pi} (\cos \omega)^k \mathbf{1}_{\{\cos \omega > 0\}} \tag{26a}$$

$$g_k(\omega) = \frac{k}{2\pi} (\cos \omega)^{k-1} \mathbf{1}_{\{\cos \omega > 0\}}, \tag{26b}$$

where $\mathbf{1}_{\{\cos \omega > 0\}}$ is an indicator function that is one if $\cos \omega > 0$ and vanishes otherwise (*Appendix 1—figure 2*). In this work, we have chosen the kernels *Equation 22a* and *Equation 22b* with $f = f_k$ and $g = g_k$ for $k = 6$. For these kernels derived here, densities $\rho(\mathbf{r}, t)$ and associated fluxes $\mathbf{J}(\mathbf{r}, t)$ that are coarse-grained on a unit sphere can be converted into effective densities and fluxes on a spherical surface of radius $R_s$ through the rescaling $\rho \to \rho/R_s^2$ and $\mathbf{J} \to \mathbf{J}/R_s$. Equivalently, rescaled kernels $K(\mathbf{r}, \mathbf{r}') \to K(\mathbf{r}, \mathbf{r}')/R_s^2$ and $\mathcal{K}(\mathbf{r}, \mathbf{r}')_{ij'} \to \mathcal{K}(\mathbf{r}, \mathbf{r}')_{ij'}/R_s$ can be used directly, as was done in *Equation 2a* and *Equation 2b* of the main text to generate the data shown in *Figure 1* (main text).

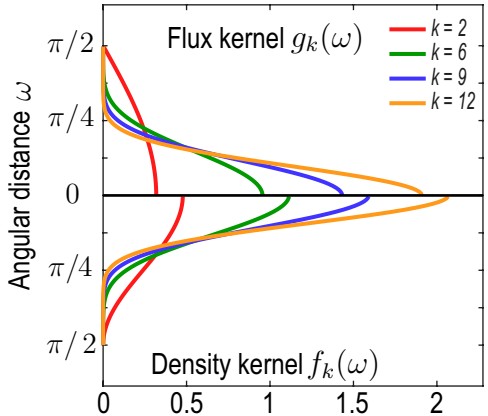

**Appendix 1—figure 2.** Family of kernel functions $f_k(\omega)$ and $g_k(\omega)$ given in *Equation 26a*. These functions represent weights of the coarse-graining kernels *Equation 22a* and *Equation 22b* that are defined such that the kernels satisfy the consistency relation *Equation 19*. $\omega = \mathrm{acos}(\mathbf{r} \cdot \mathbf{r}')$ denotes angular distances between $\mathbf{r}$ and $\mathbf{r}'$ Coarse-graining of a conserved number of particles on a sphere to determine a density field $\rho$ (*Equation 2a*, main text) requires a different weighting – $f_k(\omega)$ – than the coarse-graining of an associated flux $\mathbf{J}$ (*Equation 2b*, main text), which requires a weighting $g_k(\omega)$ instead to ensure that coarse-grained fields obey mass conservation *Equation 18*. A characteristic coarse-graining length scale associated with these kernels is the half-width at half-maximum (HWHM), which is related to $k$ by HWHM $= \arccos(2^{-1/k})$.

## Appendix 2

### Spatio-temporal mode decomposition

In this section, we provide explicit expressions for the scalar and spherical harmonic basis functions ('spatial modes'), as well as for the Chebyshev basis functions ('temporal modes') used in this work. Additionally, we describe a systematic approach to determine the minimal number of modes needed to describe the coarse-grained data, while preserving a high level of accuracy in the representation.

#### Spatial basis: spherical harmonics

In this work, we use the real spherical harmonics defined in spherical coordinates $(\theta, \phi)$ by *Arfken et al., 2013* as

$$Y_{lm}(\theta, \phi) = \sqrt{\frac{2l+1}{4\pi} \frac{(l-|m|)!}{(l+|m|)!}} P_l^{|m|}(\cos \theta) N_m(\phi) , \tag{27}$$

where $P_l^{|m|}(x)$ is the associated Legendre polynomial of degree $l$ and order $|m|$, and

$$N_m(\phi) = \begin{cases} \sqrt{2} \cos(m\phi) & \text{if } m > 0 \\ 1 & \text{if } m = 0 \\ \sqrt{2} \sin(|m|\phi) & \text{if } m < 0 \end{cases} \tag{28}$$

*Vector* spherical harmonics can be defined and expressed as vector fields in 3D or covariantly as (*Sandberg, 1978*; *Mietke et al., 2019*)

$$\mathbf{\Psi}_{lm} = \nabla_S Y_{lm} \Leftrightarrow \Psi_{(lm)}^i = g^{ij}\partial_j Y_{lm} \tag{29a}$$

$$\mathbf{\Phi}_{lm} = \hat{\mathbf{r}} \times \mathbf{\Psi}_{lm} \Leftrightarrow \Phi_{(lm)}^i = \epsilon^{ji}\partial_j Y_{lm} \tag{29b}$$

where $\nabla_S = \mathbf{e}_\theta \partial_\theta + \mathbf{e}_\phi \sin^{-1}\theta \partial_\phi$ denotes the gradient operator an a unit sphere, $\epsilon_{ij}$ is the covariant Levi-Civita tensor, and $g_{ij}$ the metric tensor. Scalar harmonics $Y_{lm}$ and either vector harmonic $\mathbf{\Lambda}_{lm} \in \{\mathbf{\Psi}_{lm}, \mathbf{\Phi}_{lm}\}$ are orthogonal:

$$\int d\Omega \, Y_{lm} Y_{l'm'} = \delta_{ll'}\delta_{mm'} \tag{30a}$$

$$\int d\Omega \, \mathbf{\Lambda}_{lm} \cdot \mathbf{\Lambda}_{l'm'} = l(l+1)\delta_{ll'}\delta_{mm'}, \tag{30b}$$

where $d\Omega = \sin\theta d\theta d\phi$. The increasing complexity of patterns and accuracy of reconstruction with larger $l$ is illustrated in *Appendix 2—figure 1* and *Video 2*.

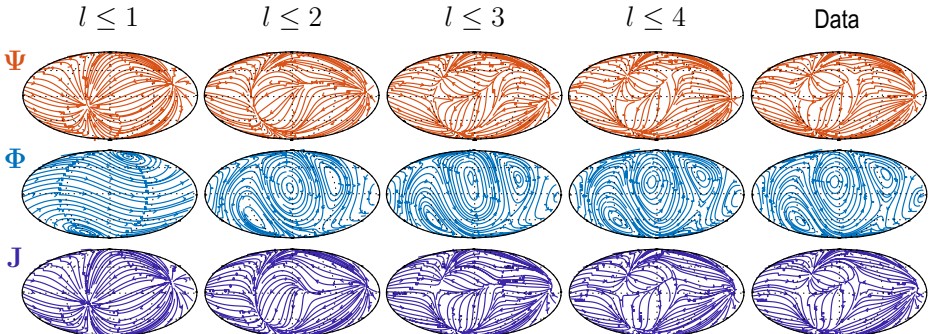

**Appendix 2—figure 1.** Sequentially adding vector spherical harmonics $\mathbf{\Psi}_{lm}$ and $\mathbf{\Phi}_{lm}$ – equivalent to increasing $l_{\max}$ in *Equation 5* – resolves increasing levels of details present in experimental flux fields ("Data"). Main features of the data are captured already by a relatively small number of modes ($l_{\max} = 4$ used throughout this work).

## Temporal basis: Chebyshev polynomials

Chebyshev polynomials of the first kind $T_n$ are defined by *Arfken et al., 2013*.

$$T_n(\cos x) = \cos(nx). \tag{31}$$

Chebyshev polynomials form an orthogonal basis of continuous functions on the interval $[-1, 1]$, such that an expansion

$$f(t) = \sum_{n=0}^{n_{\max}} c_n T_n(t) \tag{32}$$

uniformly converges as $n_{\max} \to \infty$ (*Driscoll et al., 2014*). This representation also allows computing derivatives spectrally from

$$f'(t) = \sum_{n=0}^{n_{\max}} c_n T'_n(t). \tag{33}$$

## Information loss through coarse-graining

Coarse-graining microscopic data into smooth fields is an irreversible operation, during which some of the original particle information is irretrievably lost. The choice of coarse-graining scale is thus dictated by a trade-off between smoothness and information content - choosing larger coarse-graining scales leads to smoother fields but blurs finer scale structures which may be of interest. To inform our choice of coarse-graining scale, we quantify the loss of information incurred by the coarse-graining operation.

The measure we introduce to quantify information loss is based on the the well-known relationship between the smoothness of functions in real space and Fourier space (*Stein and Shakarchi, 2011*): A smooth function in real space should have a peaked, quickly decaying spectrum in Fourier space while a collection of point-like objects such as delta functions should have a uniform non-decaying spectrum. Specifically, we describe a uniformly sampled field as a $M \times N$ matrix with components being the field values $X_{i,j} = X(\theta_i, \phi_j)$. In our case, $X_{i,j}$ represents either the density field $\rho$ or any of the Cartesian components of the flux vector field $\mathbf{J}$ at a given time point. We find the complex discrete Fourier spectrum $\hat{X}_{i,j}$ of this matrix using the two-dimensional fast Fourier transform. We then calculate the power spectral density (PSD) of the Fourier spectrum as $R_{i,j} = |\hat{X}_{i,j}|^2$ and interpret the normalized PSD.

$$P_{i,j} = \frac{R_{i,j}}{\sum_{a,b} R_{a,b}}$$

as a discrete probability distribution. The spectral entropy $S$ characterizing the information content of the field $X$ is then defined by

$$S(X) = -\frac{1}{\log_2 NM} \sum_{i,j} P_{i,j} \log_2 P_{i,j}. \tag{34}$$

Smooth fields are sharply peaked in Fourier space and have a low spectral entropy, whereas fields that resolve discrete single particle information are rather flat in Fourier space and have a large spectral entropy. The difference in entropy between particle data and smoothed fields then measures the information eliminated by the coarse-graining procedure. If we additionally normalize by the entropy of the spectral entropy $S_0(X)$ of the raw particle data, we finally obtain a relative measure of the information that is lost in the coarse-graining process. In general, a measure as given in *Equation 34* can be defined for any transform with the property that smoothness in real space leads to a fast decaying spectrum in transform space.

We compute the spectral entropy of density and flux component fields at a representative time point and for varying coarse-graining length scales (*Appendix 2—figure 2*). Specifically, we coarse-grain density and flux through the procedure described in the main text and in Appendix 1 for different values of the kernel parameter $k$ (see *Equation 26a*). Large values of $k$ correspond to small coarse-graining length scales, with the effective half-width at half-maximum (HWHM) of the kernels given in *Equation 22a* and *Equation 22b* with weight functions *Equation 26a* and *Equation 26b* scaling as $\text{HWHM} = \arccos(2^{-1/k})$. Normalized spectral entropies $S(X)/S_0(X)$ with $X \in \{\rho, \mathbf{J}\}$ are then computed using *Equation 34*. For the flux field, we define $S(\mathbf{J}) := S(J_x) + S(J_y) + S(J_z)$ ("Flux sum" in

*Appendix 2—figure 2*) and interpret the sum of these three contributions ("Flux x", "Flux y", "Flux z" in *Appendix 2—figure 2*) as the total information contained in the flux field. We find that the spectral entropies of all fields show similar features. In particular, an increasing coarse-graining width first results in a sharp loss of information as individual particle positions are blurred, followed by less steep information loss as continuous fields progressively lose details of finer structures. In this work, we use an intermediate value of the coarse-graining parameter $k = 6$ (yellow data in *Appendix 2—figure 2*).

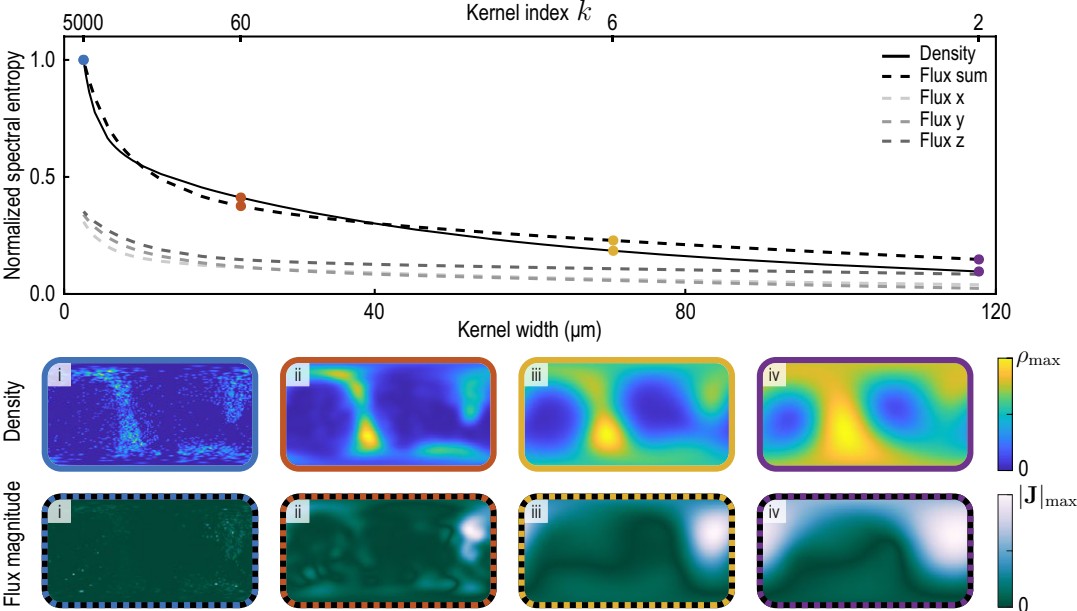

**Appendix 2—figure 2.** Normalized spectral entropy as a function of the coarse-graining kernel width (top) computed for density $\rho$ and flux field $\mathbf{J}$ using *Equation 34*. To evaluate the spectral entropy for the vector-valued flux, we define $S(\mathbf{J}) := S(J_x) + S(J_y) + S(J_z)$ ('Flux sum'). The coarse-graining width – the half-width at half-maximum (HWHM) of the coarse-graining kernels *Equation 22a* and *Equation 22b* with weight functions *Equation 26a* – is varied by varying the kernel index $k$, where $\mathrm{HWHM} = \arccos(2^{-1/k})$ (see *Appendix 1—figure 2*). The fields $\rho$ and $|\mathbf{J}|$ are shown in the two bottom rows for different values of $k$ i. $k = 5000$ (blue, data used to compute the reference spectral entropies $S_0(\rho)$ and $S_0(\mathbf{J})$) ii. $k = 60$ (brown) iii. $k = 6$ (yellow, used in main text) iv. $k = 2$ (purple).

## Optimal compression in space and time

Spectral representations are exact in the limit of an infinite number of modes. In practice, we choose a maximal harmonic mode number $l_{\max}$ and maximal Chebyshev mode number $n_{\max}$. A too large value of $l_{\max}$ and $n_{\max}$ provides little compression benefit, while too small values suffer accuracy penalties. Hence, there is a compression-accuracy trade-off that we seek to optimize. To evaluate the trade-off quantitatively, we define a heuristic compression metric $C$ by

$$1/C = \frac{n_{\max}}{N_t} + \frac{(l_{\max}+1)^2}{N_s}, \tag{35}$$

where $N_t$ is the number of sampled time steps and $N_s$ is the number of spatial grid points used for coarse-graining. Larger values of $C$ correspond to a higher compression factor. To define accuracy metrics, we consider the norm

$$\|f\|^2 = \sum_{i=1}^{N_t} f(t_i)^2$$

where the sum runs over $N_t$ regularly sampled time points $t_i$. We denote a particular mode representation $\{\tilde{\rho}_{lm}(t), \tilde{j}_{lm}^{(1)}(t), \tilde{j}_{lm}^{(2)}(t)\}$ of the data that was coarse-grained via *Equation 2a* and *Equation*

**2b** (main text) for $l = 0, \ldots, l_{\max}^{\text{ref}} = 20$ as the 'uncompressed' reference. A measure to characterize the accuracy of a mode-truncated 'compressed' data representation is then given by a relative average mode reconstruction error

$$E_{\text{modes}}(n_{\max}, l_{\max}) = \frac{1}{2(l_{\max}^{\text{ref}}+1)^2} \sum_{l=0}^{l_{\max}} \sum_{m=-l}^{m=l} \left( \frac{||\rho_{lm} - \tilde{\rho}_{lm}||^2}{||\tilde{\rho}_{lm}||^2} + \frac{||j_{lm}^{(2)} - \tilde{j}_{lm}^{(2)}||^2}{||\tilde{j}_{lm}^{(2)}||^2} \right)^{1/2}. \tag{36}$$

This measure compares the compressed mode representation $\{\rho_{lm}(t), j_{lm}^{(2)}(t)\}$, truncated at maximal Chebychev mode number $n_{\max}$ (temporal representation **Equation 32**, Appendix 2) and maximal harmonic mode number $l_{\max}$ (spatial representation, **Equations 4; 5**, main text) with the reference modes $\{\tilde{\rho}_{lm}(t), \tilde{j}_{lm}^{(2)}(t)\}$. To find a compromise between accuracy, characterized by $E_{\text{modes}}(n_{\max}, l_{\max})$, and compression $C$ defined in **Equation 35**, the aim is to find a pair $(n_{\max}, l_{\max})$ on the Pareto front (**Jin and Sendhoff, 2008**) of $E_{\text{modes}}$ vs. $1/C$ (red dots in **Appendix 2—figure 3**).

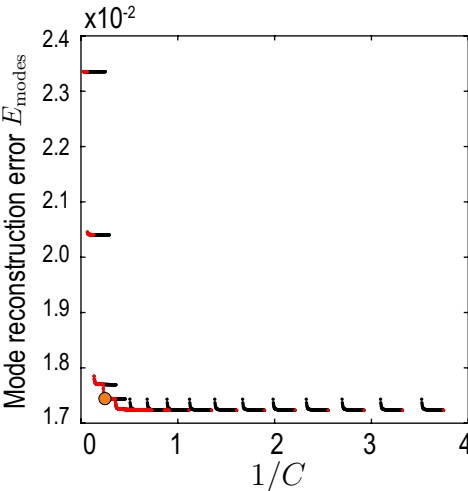

**Appendix 2—figure 3.** Relative average mode reconstruction error $E_{\text{modes}}(n_{\max}, l_{\max})$ **Equation 36**. As a function of the inverse of the compression $C$ defined in **Equation 35**. Red points indicate the Pareto front (**Jin and Sendhoff, 2008**) of this compression-accuracy approximation trade-off. Orange circle indicates the final value used for our analysis.

Note that the modes $\tilde{j}_{lm}^{(1)}(t)$ and $j_{lm}^{(1)}(t)$ are so far omitted from this analysis, because the latter are in practice found directly from density modes via **Equation 6** (main text). However, taking temporal derivatives of $\rho_{lm}(t)$ using **Equation 33** to determine $j_{lm}^{(1)}(t)$ introduces undesirable oscillations for too large Chebychev cut-offs $n_{\max}$. This implies an additional trade-off between the need for accuracy (higher $n_{\max}$) and stability (lower $n_{\max}$). In practice, we wish to find values of $(n_{\max}, l_{\max})$ such that relative amplitudes of pairs $(\tilde{j}_{lm}^{(1)}, \tilde{j}_{lm}^{(2)})$ and $(j_{lm}^{(1)}, j_{lm}^{(2)})$ are preserved by the compression. This can be achieved by comparing the relative curl amplitude

$$S_{\text{curl}}(t) = \frac{\sum_{lm}[j_{lm}^{(2)}(t)]^2}{\sum_{lm}[j_{lm}^{(1)}(t)]^2 + [j_{lm}^{(2)}(t)]^2}$$

to the analog quantity $\tilde{S}_{\text{curl}}(t)$ computed from the reference modes $\{\tilde{j}_{lm}^{(1)}(t), \tilde{j}_{lm}^{(2)}(t)\}$ and analyzing the curl reconstruction error

$$E_{\text{curl}} = \frac{||S_{\text{curl}} - \tilde{S}_{\text{curl}}||}{||\tilde{S}_{\text{curl}}||} \tag{37}$$

as a function of $n_{\max}$ and $l_{\max}$ (**Appendix 2—figure 4**). From this, we find a region of low error around $l_{\max} = 4, n_{\max} = 30$, which also is on the Pareto front of the accuracy vs. compression trade-off (orange circles in **Appendix 2—figures 3 and 4**) and represents the final values used throughout this work.

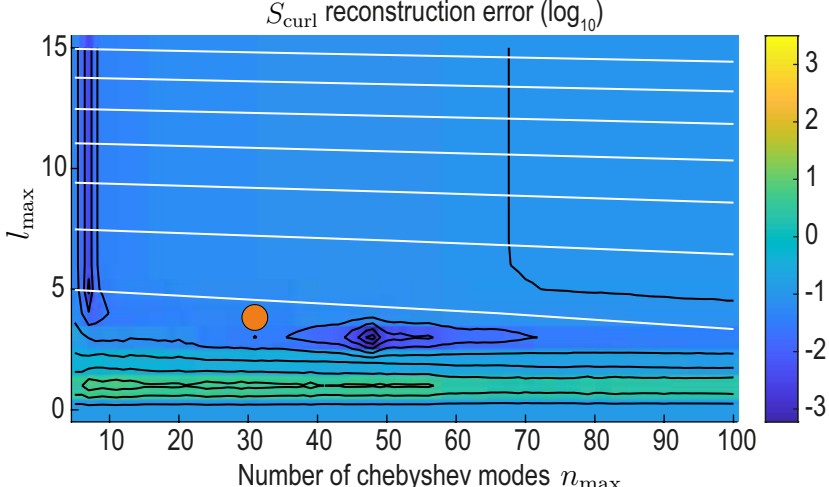

**Appendix 2—figure 4.** $S_{\mathrm{curl}}$ reconstruction error landscape (log scale) as a function of $l_{\max}$ and $n_{\max}$. Black contour lines indicate iso-error lines (see *Equation 37*, $E_{\mathrm{curl}} = $ const.), whereas white contour lines indicate iso-compression levels (see *Equation 35*, $C = $ const.). Orange circle indicates the final value used for our analysis.

## Appendix 3

### Active Brownian particles on the sphere

In this section, we describe the stochastic dynamics of non-interacting, active Brownian particles (ABPs) (*Romanczuk et al., 2012*) on curved surfaces and derive analytically coarse-grained mean-field equations, as well as a kernel representation of ABP dynamics. These results are used in the main text to validate our coarse-graining and inference framework.

We consider active Brownian particles at position $\mathbf{x} \in \mathbb{R}^3$ that move with speed $v_0$ on the surface of a unit sphere (radius $R_0 = 1$) in the direction of their unit orientation vector $\mathbf{u} \in \mathbb{R}^3$. Since $|\mathbf{x}| = 1$ at all times, we can interpret $v_0$ as the particle's angular speed on the unit sphere. The orientation vector is at all times tangential to the surface, but is subject to random in-plane fluctuations characterized by a rotational diffusion coefficient $D_r$. The corresponding dynamics of $\mathbf{x}(t)$ and $\mathbf{u}(t)$ is given by the stochastic differential equations (in units $R_0 = 1$)

$$\mathrm{d}\mathbf{x} = v_0\mathbf{u}\,\mathrm{d}t \tag{38a}$$

$$\mathrm{d}\mathbf{u} = -v_0\mathbf{x}\mathrm{d}t + \left(\mathbf{x} \times \mathbf{u}\right)\sqrt{2D_r} \circ \mathrm{d}\xi, \tag{38b}$$

where the stochastic differential *Equation 38b* is interpreted in the Stratonovich sense, as denoted by the symbol "∘" (*Braumann, 2007*). It follows from *Equation 38a* that $\mathbf{x}(t)$ and $\mathbf{u}(t)$ are normalized at all times. In the absence of rotational diffusion ($D_r = 0$), the vectors $\mathbf{x}$ and $\mathbf{u}$ rotate over time by an angle $v_0t$ around the axis $\mathbf{u} \times \mathbf{x}$. Consequently, particle trajectories in the absence of noise trace out great circles in the plane defined by $(\mathbf{u} \times \mathbf{x})$.

### Spatial correlation of APBs on a sphere

$C(t) = \langle\mathbf{x}(t) \cdot \mathbf{x}(0)\rangle$ To illustrate how ABPs on a sphere differ from ABPs in Euclidean space, we study first the correlation function , where the angled brackets denote a Gaussian white-noise average. To this end, we rewrite the ABP dynamics *Equation 38a* in their equivalent Itô form given by

$$\mathrm{d}\mathbf{x} = v_0\mathbf{u}\,\mathrm{d}t \tag{39a}$$

$$\mathrm{d}\mathbf{u} = -\left(v_0\mathbf{x} + D_r\mathbf{u}\right)\mathrm{d}t + \sqrt{2D_r}\left(\mathbf{x} \times \mathbf{u}\right)\mathrm{d}\xi. \tag{39b}$$

In the Itô formulation any smooth function $f(\mathbf{x}, \mathbf{u})$ obeys $\langle f(\mathbf{x}, \mathbf{u})\mathrm{d}\xi\rangle = 0$, such that (*Winkler et al., 2015*)

$$\frac{\mathrm{d}}{\mathrm{d}t}\langle\mathbf{x}(t) \cdot \mathbf{x}(0)\rangle = v_0\langle\mathbf{u}(t) \cdot \mathbf{x}(0)\rangle$$

and

$$\frac{\mathrm{d}}{\mathrm{d}t}\langle\mathbf{u}(t) \cdot \mathbf{x}(0)\rangle = -v_0\langle\mathbf{x}(t) \cdot \mathbf{x}(0)\rangle - D_r\langle\mathbf{u}(t) \cdot \mathbf{x}(0)\rangle,$$

which yields a damped harmonic oscillator equation for the correlation function

$$\frac{\mathrm{d}^2}{\mathrm{d}t^2}C(t) + D_r\frac{\mathrm{d}}{\mathrm{d}t}C(t) + v_0^2 C(t) = 0. \tag{40}$$

Normalization and orthogonality of $\mathbf{x}(t)$ and $\mathbf{u}(t)$ imply the initial conditions $C = 1$ and $\mathrm{d}C/\mathrm{d}t = 0$ at $t = 0$. The behavior of solutions of *Equation 40* is a function of the rotational Péclet number $\mathrm{Pe}_r := v_0/D_r$ that quantifies the ratio between active motion and orientational diffusion. For $\mathrm{Pe}_r < 1$, ('high-noise regime'), the position correlation function $C(t) = \langle\mathbf{x}(t) \cdot \mathbf{x}(0)\rangle$ decays according to *Equation 40* monotonically to zero. For $\mathrm{Pe}_r > 1$, ('low -noise regime') position correlations exhibit damped oscillations. To validate our simulation method (described in the following section), analytic predictions for $C(t)$ are in *Figure 3B* (main text) compared against the ensemble average $\langle\mathbf{x}(t) \cdot \mathbf{x}(0)\rangle$ over $3 \times 10^4$ simulated ABPs.

## Stochastic simulation of active Brownian particles on the sphere

To ensure a numerically exact normalization of the particle's position and orientation vectors on the unit sphere, we simulated the dynamics

$$d\mathbf{x} = \frac{v_0}{|\mathbf{u}|}\left(\mathbf{u} - \frac{\mathbf{u}\cdot\mathbf{x}}{|\mathbf{x}|^2}\mathbf{x}\right)dt \tag{41a}$$

$$d\mathbf{u} = -v_0\frac{\mathbf{x}}{|\mathbf{x}|^2}dt + \frac{(\mathbf{x}\times\mathbf{u})}{|\mathbf{x}\times\mathbf{u}|}\sqrt{2D_r}\circ d\xi. \tag{41b}$$

We numerically solve the Itô formulation of this system using the Euler-Mayurama scheme (*Higham., 2001*), and confirm that this system reproduces the correlation dynamics predicted by *Equation 40* (*Figure 3B*, main text).

## Fokker-Planck equation

To study the continuum dynamics of a large number of non-interacting ABPs on a sphere, we determine the dynamics of the probability density $p(\mathbf{x}, \mathbf{u}, t)$ of particle positions $\mathbf{x}$ and orientations $\mathbf{u}$ at time $t$. To do so, it is convenient to express particle positions in terms of a parameterisation $\mathbf{x}(t) = \mathbf{x}[x^1(t), x^2(t)]$ that defines tangential basis vectors by $\mathbf{e}_i = \partial\mathbf{x}/\partial x^i$ $(i = 1, 2)$ and a metric tensor $g_{ij} = \mathbf{e}_i \cdot \mathbf{e}_j$. By definition, we have $d\mathbf{x} = \mathbf{e}_i dx^i$ and *Equation 38a* can be rewritten as

$$dx^i = v_0 u^i dt. \tag{42}$$

General tangential vectors on the surface can be written as $\mathbf{u} = u^i\mathbf{e}_i$ and on a unit sphere the surface normal can be identified with particle positions $\mathbf{n} = \mathbf{e}_1 \times \mathbf{e}_2/|\mathbf{e}_1 \times \mathbf{e}_2| = \mathbf{x}$. Hence, on the unit sphere the Gauss-Weingarten relation reads $\partial_i\mathbf{e}_j = -C_{ij}\mathbf{x} + \Gamma_{ij}^k\mathbf{e}_k$, where $\Gamma_{ij}^k$ denote Christoffel symbols and $C_{ij}$ is the curvature tensor. This implies together with *Equation 42* the geometric relation

$$d\mathbf{u} = \mathbf{e}_i du^i + u^i(\partial_j\mathbf{e}_i)dx^j$$
$$= \mathbf{e}_i du^i - C_{ij}u^i u^j v_0\mathbf{x}dt + v_0 u^i u^j\Gamma_{ij}^k\mathbf{e}_k dt.$$

Comparing this identity with the stochastic dynamics $d\mathbf{u}$ in *Equation 38b* and using that $C_{ij}u^i u^j = g_{ij}u^i u^j = |\mathbf{u}|^2 = 1$ for unit vectors $\mathbf{u}$ on the unit sphere, we find the covariant stochastic differential equation

$$du^i = -v_0 u^j u^k\Gamma_{jk}^i dt + \epsilon^i_{\ k}u^k\sqrt{2D_r}\circ d\xi. \tag{43}$$

In *Equation 43*, $\epsilon_{ij} = \mathbf{x}\cdot(\mathbf{e}_i \times \mathbf{e}_j)$ denotes the Levi-Civita tensor on the unit sphere.
In this covariant basis, we define the scalar probability density

$$p(\mathbf{x}, \mathbf{u}, t) = \left\langle \frac{1}{\sqrt{g(\mathbf{x})}}\prod_i \delta[x^i - x^i(t)]\delta[u^i - u^i(t)]\right\rangle, \tag{44}$$

where $\delta(x)$ denotes a Dirac function. Combining *Equations 42; 43*, standard methods (*Fily et al., 2016*; *Castro-Villarreal and Sevilla, 2018*) allow us to obtain the Fokker-Planck equation for $p(\mathbf{x}, \mathbf{u}, t)$ as

$$\frac{\partial}{\partial t}p(\mathbf{x}, \mathbf{u}, t) = D_r\frac{\partial}{\partial u^i}\left[\epsilon^i_{\ k}u^k\frac{\partial}{\partial u^j}\left(\epsilon^j_{\ l}u^l p\right)\right] - \nabla_i(v_0 u^i p) + \frac{\partial}{\partial u^i}\left(v_0 u^j u^k\Gamma_{jk}^i p\right) \tag{45}$$

Using the identity $\epsilon^i_{\ k}\epsilon^j_{\ l} = g^{ij}g_{kl} - \delta^i_l\delta^j_k$, the dynamics of the probability density is finally given by

$$\frac{\partial}{\partial t}p(\mathbf{x}, \mathbf{u}, t) = D_r\frac{\partial}{\partial u^i}\left[(g^{ij} - u^i u^j)\frac{\partial p}{\partial u^j}\right] - v_0 u^i\nabla_i p + \frac{\partial}{\partial u^i}\left(v_0 u^j u^k\Gamma_{jk}^i p\right), \tag{46}$$

which agrees with the result in *Castro-Villarreal and Sevilla, 2018*.

## Hydrodynamic expansion

To connect the Fokker-Planck dynamics given in *Equation 46* to hydrodynamic fields, we define (probability) density and fluxes by $\rho(\mathbf{x}, t) = \int d^2\mathbf{u}\, p(\mathbf{x}, \mathbf{u}, t)$, and $J^i(\mathbf{x}, t) = v_0\int d^2\mathbf{u}\, u^i p(\mathbf{x}, \mathbf{u}, t)$. Their dynamics on the unit sphere is given by *Castro-Villarreal and Sevilla, 2018*.

$$\frac{\partial \rho}{\partial t} = -\nabla_i J^i \tag{47a}$$

$$\frac{\partial J^i}{\partial t} = -\frac{v_0^2}{2} \nabla^i \rho - D_r J^i, \tag{47b}$$

where couplings to higher order fields are neglected, as they vanish at shorter time-scales due to the presence of rotational noise. Expressing *Equation 47a* and *Equation 47b* in terms of scalar and vector spherical harmonics (see Appendix 2) for an arbitrary sphere radius $R_0$ yields the mode dynamics *Equation 13a*, *Equation 13b* and *Equation 13c* the main text.

## Appendix 4

### Learning and interpreting the linear model

We describe details about the inference procedure used to learn the linear ordinary differential equation (ODE) model considered in the main text. We then discuss how the matrix $M$ found by this procedure can be further studied in terms of its real-space kernel representation and derive this kernel for the ABP dynamics introduced in Appendix 4.

#### Inference of the dynamical mode coupling matrix $M$

Given a dynamical mode vector $\mathbf{a}(t) = \left[ \rho_{lm}(t), j_{lm}^{(1)}(t), j_{lm}^{(2)}(t) \right]^\top$, the goal is to learn a linear minimal model

$$\frac{\mathrm{d}\mathbf{a}(t)}{\mathrm{d}t} = M \cdot \mathbf{a}(t) \tag{48}$$

of the mode dynamics. Here, $M$ is an unknown $n \times n$ mode coupling matrix, where generally $n = 3(l_{\max} + 1)^2 - 2$. In systems with global mass conservation, as considered in this work, one can additionally use that the mode $\rho_{00}$ is constant and eliminate the corresponding couplings from $M$.

To describe the algorithm that was used to infer the mode coupling matrix $M$, we parameterize $M$ by a vector $\mathbf{p}$ that contains all non-zero entries and introduce a function $\mathcal{M}$ that represents the underlying matrix structure. Together, they generate the explicit form $M = \mathcal{M}(\mathbf{p})$ of the mode coupling matrix. Imposing structure on the matrix, such as rank constraints, or sparsity leads to a shorter vector $\mathbf{p}$ and modifies the definition of $\mathcal{M}$ accordingly. Denoting $\mathbf{A}(t; \mathcal{M}, \mathbf{p}, \mathbf{a}_0)$ as the result of numerically integrating the system of ODEs *Equation 48* up to time $t$ from initial condition $\mathbf{a}_0$ with $M = \mathcal{M}(\mathbf{p})$, we define the loss function

$$L(\mathbf{p}; \mathcal{M}, t_I, t_N) = \frac{1}{N-I} \sum_{i=I}^{N} \|\mathbf{a}(t_i) - \mathbf{A}(t_i; \mathcal{M}, \mathbf{p}, \mathbf{a}(t_I))\|_2^2, \tag{49}$$

where the $t_i$ are time points in an interval $[t_I, t_N]$ at which the data and the ODE solution are sampled. Using the ODE solvers and optimization functions provided by the Julia modules DifferentialEquations.jl and DiffEqFlux.jl (*Rackauckas et al., 2021*), we can differentiate through the ODE solver to calculate derivatives of the loss function *Equation 49* with respect to parameters $\mathbf{p}$ and subsequently apply gradient-based optimization algorithms. The loss function is minimized using the ADAM algorithm (*Kingma and Ba, 2017*), followed by the Broyden-Fletcher-Goldfarb-Shannon (BFGS) algorithm (*Nocedal and Wright, 2006*). To increase the robustness of the optimization and promote sparsity, we use a sequentially thresholded algorithm (*Supekar et al., 2021*; *Brunton et al., 2016*; *Reinbold, 2020*). A complete overview of this procedure is shown in *Appendix 4—figure 1* and the details of the specific design decisions made in the algorithm are discussed in the following:

1. To account for the variation in scale between the different modes in the data $\mathbf{a}(t)$, each mode is normalized by its median absolute deviation (MAD) across the full time-span in which the data are available. Specifically, we scale each mode by

$$\mathrm{mad}(a_i) = \mathrm{median}_k \left( |a_i(t_k) - \bar{a}_i| \right). \tag{50}$$

where $\bar{a}_i = \mathrm{median}_k[a_i(t_k)]$ and the median is taken over all time-points, giving rise to a scaled mode vector $\tilde{\mathbf{a}}(t)$. Losses analogous to *Equation 49* that are computed using scaled data are denoted in the following by $\tilde{L}$.

2. To prevent over-fitting, we divide the data into two regions, a learning region from $t_I$ to $t_N$ and a validation region from $t_N$ to $t_F$. Only data from the learning region is used in the optimization of the loss function *Equation 49*. However, the model is integrated into the validation region, and a corresponding validation loss using only the data in the validation region is calculated. During each optimization run, we choose the model with the lowest loss in the validation region, lowering the likelihood of over-fitting to the specific data in the learning region.

3. To prevent the optimization from getting stuck in local minima, we incrementally increase the time-span of the data included in the optimization objective (blue box in *Appendix 4—figure 1*). We increase the time window backward from a fixed endpoint $t_1 = t_F$, choosing each iteration an

earlier initial condition at time $t_i < t_{i-1}$. The advantage of stepping backward rather than forward from a fixed initial condition is twofold: first, the validation region stays unchanged throughout the optimization, making comparisons of the validation loss easy. Second, because the initial condition changes with each run, the learned matrix tends to be more robust to fluctuations in the initial condition.

4. After the optimization step, sparsity is promoted by thresholding the elements in the matrix (*Brunton et al., 2016*), removing small magnitude elements that do not noticeably contribute to the mode dynamics (purple box in *Appendix 4—figure 1*). The optimization procedure is then repeated until the thresholding converges. The threshold is chosen to generate a sparse matrix that still reproduces the dynamics faithfully.

5. Once the sparsity pattern is obtained from the sequential thresholding and optimization procedure a final run of the optimization is performed on the unscaled mode data to find the final dynamical matrix $M$, which removes any potential slight bias the MAD scaling might have introduced in the parameter values **p**.

Finally, the numerical stability of the model can be checked by examining the eigenvalues of the learned matrix. For the ABP test data, we learn a matrix $M$ for which the largest real part of its eigenvalues is at machine precision. For the experimental data, the largest real part in the eigenvalues is $7.4 \times 10^{-4}$, which corresponds to a time scale of around 675 mins. While the corresponding dynamics will eventually become unstable, solutions remain bound over a period of approximately 45 hours, which is four times as long as the input data from which the mode coupling matrix was learned.

## Learning and validation regions used in this work

For the ABP data, the first 15 frames are excluded, so that – consistent with coarse-graining assumptions (see Appendix 3, *Equation 47a*, *Equation 47b*) any remnants of higher orientational order introduced by the initial conditions have decayed. The subsequent 140 frames are used as the learning region, followed by a validation region of 20 frames. Each frame corresponds to a time interval of approximately 0.06 in units of $R_0/v_0 = 1$. We exclude the first and last 10 frames of the experimental zebrafish data and split the remaining data into a learning region of 360 frames, with the remaining 40 frames used for validation. Each frame corresponds to a time interval of 2 min.

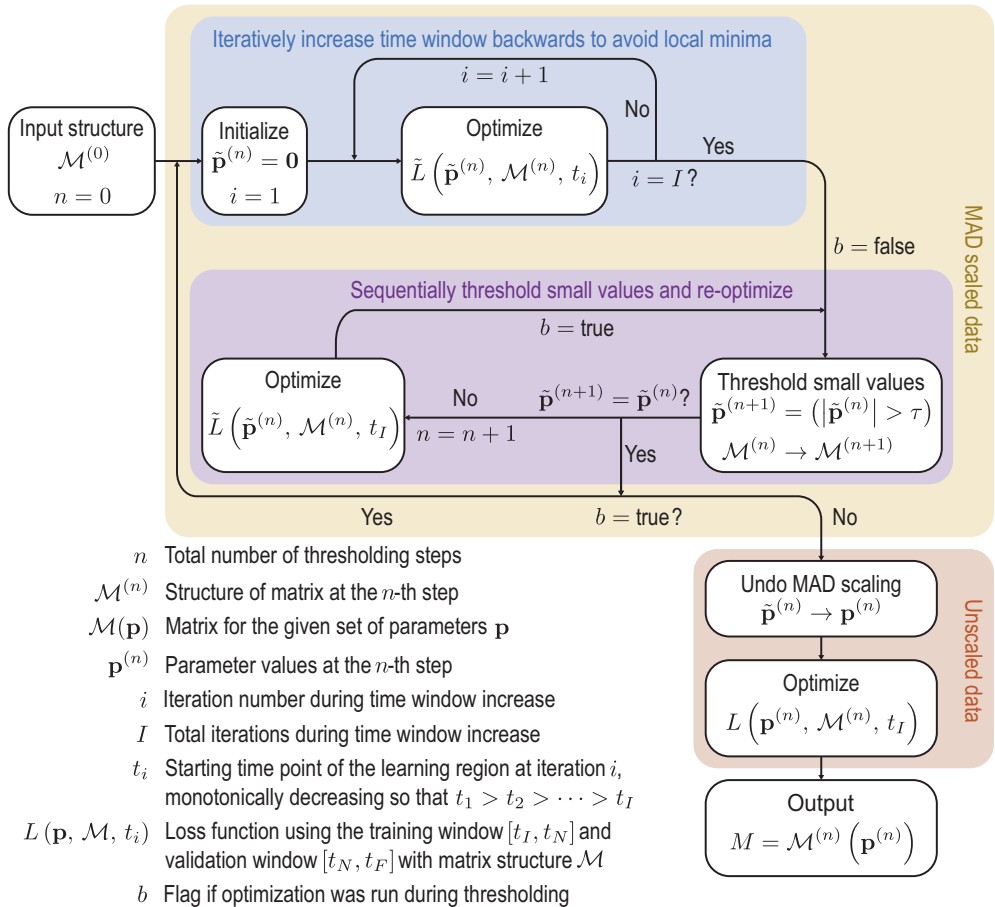

**Appendix 4—figure 1.** Initially the data is rescaled using the median absolute deviation (MAD) defined in *Equation 50* to account for variation in scales across the modes. Scaled variables are denoted by tildes. To avoid local minima of the optimization function, we iteratively feed more data into the cost function. Next we sequentially threshold the small terms in the matrix until convergence is reached. These procedures are repeated until the sparsity pattern converges. Finally the scaling is undone and the parameters are optimized on the unscaled data to produce the final matrix. Schematic of the learning procedure.

## Green's function representation of the learned matrix

The learned matrix $M$ consists of 9 blocks each with $[(l_{max} + 1)^2 - 1] \times [(l_{max} + 1)^2 - 1]$ entries. Each block relates a mode family to time derivatives of another and we write

$$
\mathbf{M} = \left(
\begin{array}{c|c|c}
M^{\rho\rho} & M^{\rho 1} & M^{\rho 2} \\
\hline
M^{1\rho} & M^{11} & M^{12} \\
\hline
M^{2\rho} & M^{21} & M^{22}
\end{array}
\right).
$$

We denote the components of each block by $\left(M^{m_1 m_2}\right)_{lm,l'm'} \equiv M^{m_1 m_2}_{\alpha\beta}$, where $m_1, m_2 \in \{\rho, 1, 2\}$, and $\alpha$, $\beta$ are multi-indices that represent the harmonic modes $(lm)$. Using the mode representation *Equation 5* and the form of the linear minimal model *Equation 48*, we find

$$
\begin{aligned}
\frac{\partial}{\partial t} \mathbf{J}(\mathbf{r}, t) &= \sum_{\alpha=lm} \left( \frac{d j^{(1)}_\alpha(t)}{dt} \boldsymbol{\Psi}_\alpha(\hat{\mathbf{r}}) + \frac{d j^{(2)}_\alpha(t)}{dt} \boldsymbol{\Phi}_\alpha(\hat{\mathbf{r}}) \right) \\
&= \sum_{\alpha=lm} \sum_{\beta=l'm'} \left[ M^{1\rho}_{\alpha\beta} \rho_\beta(t) + M^{11}_{\alpha\beta} j^{(1)}_\beta(t) + M^{12}_{\alpha\beta} j^{(2)}_\beta(t) \right] \boldsymbol{\Psi}_\alpha(\hat{\mathbf{r}}) \\
&\quad + \left[ M^{2\rho}_{\alpha\beta} \rho_\beta(t) + M^{21}_{\alpha\beta} j^{(1)}_\beta(t) + M^{22}_{\alpha\beta} j^{(2)}_\beta(t) \right] \boldsymbol{\Phi}_\alpha(\hat{\mathbf{r}}).
\end{aligned}
\tag{51}
$$

Using *Equation 30a*, *Equation 51* can be cast into the dynamic kernel *Equation 14* given in the main text, where we defined the vector kernel

$$\mathbf{m}^{\rho}(\mathbf{r}, \mathbf{r}') = \sum_{\alpha=lm} \sum_{\beta=l'm'} M^{1\rho}_{\alpha\beta} \boldsymbol{\Psi}_{\alpha}(\hat{\mathbf{r}}) Y_{\beta}(\hat{\mathbf{r}}') + M^{2\rho}_{\alpha\beta} \boldsymbol{\Phi}_{\alpha}(\hat{\mathbf{r}}) Y_{\beta}(\hat{\mathbf{r}}') \tag{52}$$

and the matrix kernel

$$\begin{aligned} M^{J}(\mathbf{r}, \mathbf{r}') \quad &= \sum_{\alpha=lm} \sum_{\beta=l'm'} \frac{1}{l(l+1)} \Big[ M^{11}_{\alpha\beta} \boldsymbol{\Psi}_{\alpha}(\hat{\mathbf{r}}) \otimes \boldsymbol{\Psi}_{\beta}(\hat{\mathbf{r}}') \\ &+ M^{12}_{\alpha\beta} \boldsymbol{\Psi}_{\alpha}(\hat{\mathbf{r}}) \otimes \boldsymbol{\Phi}_{\beta}(\hat{\mathbf{r}}') + M^{21}_{\alpha\beta} \boldsymbol{\Phi}_{\alpha}(\hat{\mathbf{r}}) \otimes \boldsymbol{\Psi}_{\beta}(\hat{\mathbf{r}}') \\ &+ M^{22}_{\alpha\beta} \boldsymbol{\Phi}_{\alpha}(\hat{\mathbf{r}}) \otimes \boldsymbol{\Phi}_{\beta}(\hat{\mathbf{r}}') \Big], \end{aligned} \tag{53}$$

where $\otimes$ denotes a dyadic product. The matrix $M^{J}(\mathbf{r}, \mathbf{r}')$ has a 0 eigenvalue with right eigenvector $\hat{\mathbf{r}}'$ and left eigenvector $\hat{\mathbf{r}}$, which implies $\det\left(M^{J}\right) = 0$. Numerical analysis of the matrix invariants shows that a second eigenvalue is 0 (*Appendix 4—figure 2*), leaving only a single non-zero eigenvalue that can be conveniently found from $\mathrm{tr}\left[M^{J}(\mathbf{r}, \mathbf{r}')\right]$ and is shown in *Figure 4D* (main text).

## Real-space kernels of active Brownian particle dynamics

In the following we determine a real-space kernel representation in the form *Equation 14* for the flux dynamics of ABPs given in *Equation 47b*. We can read off the kernel coefficients in *Equation 52* and in *Equation 53* from the coarse-grained ABP dynamics in mode space, given in *Equation 13b* and *Equation 13c*. For the kernel $\mathbf{m}^{\rho}(\mathbf{r}, \mathbf{r}')$, we have $M^{1\rho}_{\alpha\beta} = -\frac{v_0^2}{2}\delta_{\alpha\beta}$ and $M^{2\rho}_{\alpha\beta} = 0$ ($\alpha, \beta = (lm)$), such that *Equation 52* becomes

$$\mathbf{m}^{\rho}(\mathbf{r}, \mathbf{r}') = -\frac{v_0^2}{2} \nabla_{\mathcal{S}} \sum_{\alpha=lm} Y_{\alpha}(\hat{\mathbf{r}}) Y_{\alpha}(\hat{\mathbf{r}}') = -\frac{v_0^2}{2} \nabla_{\mathcal{S}} \delta(\mathbf{r} - \mathbf{r}'). \tag{54}$$

Here, we have used in the first step the definition of $\boldsymbol{\Psi}_{lm}(\hat{\mathbf{r}})$ given in *Equation 29a* and in the second step the completeness of the spherical harmonic basis functions $Y_{lm}(\hat{\mathbf{r}})$, where $\delta(\mathbf{r} - \mathbf{r}') = \delta(\phi - \phi')\delta(\cos\theta - \cos\theta')$ denotes the delta function on a sphere. Note that a unit sphere was considered throughout this analysis, such that $\mathbf{r} = \hat{\mathbf{r}}$. Similarly, *Equation 13b* and *Equation 13c* imply for the kernel coefficients in *Equation 53* that $M^{11}_{\alpha\beta} = M^{22}_{\alpha\beta} = -D_r\delta_{\alpha\beta}$ and $M^{12}_{\alpha\beta} = M^{21}_{\alpha\beta} = 0$. Consequently, we have

$$M^{J}(\mathbf{r}, \mathbf{r}') = -D_r \sum_{\alpha=lm} \frac{1}{l(l+1)} \left[ \boldsymbol{\Psi}_{\alpha}(\hat{\mathbf{r}}) \otimes \boldsymbol{\Psi}_{\alpha}(\hat{\mathbf{r}}') + \boldsymbol{\Phi}_{\alpha}(\hat{\mathbf{r}}) \otimes \boldsymbol{\Phi}_{\alpha}(\hat{\mathbf{r}}') \right] = -D_r\delta(\mathbf{r} - \mathbf{r}')P_{\parallel}, \tag{55}$$

where $P_{\parallel} = \mathbb{I} - \mathbf{r} \otimes \mathbf{r}$ is the tangential projector on the unit sphere. The hydrodynamic flux *Equation 47b* of ABPs on a sphere can therefore be written in the equivalent integral kernel form

$$\partial_t \mathbf{J}(\mathbf{r}, t) = \int \mathrm{d}\Omega' \left[ -\frac{v_0^2}{2} \nabla_{\mathcal{S}} \delta(\mathbf{r} - \mathbf{r}')\rho(\mathbf{r}', t) - D_r\delta(\mathbf{r} - \mathbf{r}')\mathbf{J}(\mathbf{r}', t) \right]. \tag{56}$$

To make analytic kernel properties comparable to practical inference scenarios in which we work with a finite number of harmonic modes, we computed the sums in *Equations 54; 55* up to a maximum mode number $l_{\max} = 4$. The resulting kernels – depicted in *Figure 4D* (main text) – approximate the Dirac delta function $\delta(\mathbf{r} - \mathbf{r}')$ and its derivative, leading to the finite range of $\mathrm{tr}(M^{J})$ with amplitude maximum at $\omega = 0$, while $|\mathbf{m}^{\rho}|$ vanishes at and peaks away from $\omega = 0$. Additionally, finite mode representations introduce an apparent kernel inhomogeneity across the spherical surface as evident from the non-zero standard deviation depicted in *Figure 4D* of the main text (blue shades).

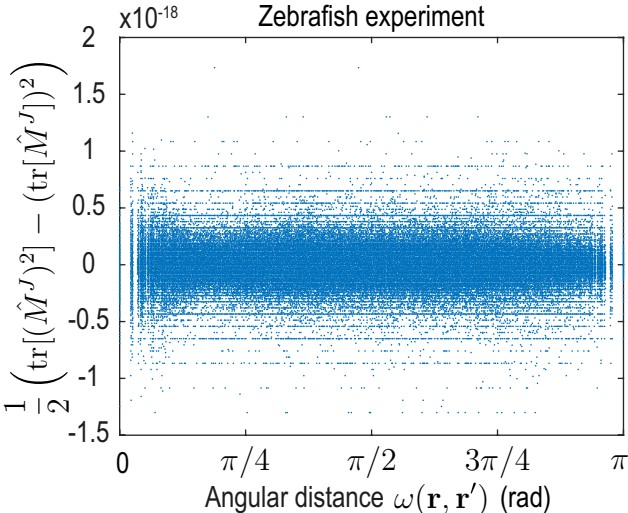

**Appendix 4—figure 2.** The $3 \times 3$-matrix invariant $I_2 = \frac{1}{2}\left(\text{tr}[(M^J)^2] - (\text{tr}[M^J])^2\right)$ sampled for pairs of positions $\mathbf{r}$, $\mathbf{r}'$ vanishes to machine precision for the dynamical matrix $M$ learned on the zebrafish data. This invariant can be expressed in terms of matrix eigenvalues as $I_2 = \lambda_1\lambda_2 + \lambda_1\lambda_3 + \lambda_2\lambda_3$. Additionally, $\lambda_1\lambda_2\lambda_3 = \det(M^J) = 0$ (see discussion below *Equation 53*), which implies only one eigenvalue is non-zero. Evaluating $I_2$ for the kernel matrix $M^J$ encoded by the theoretical (see *Equation 13b* and *Equation 13c*) and inferred (see *Figure 3*C, main text) dynamical matrix $M$ of the ABP dynamics yields similar results.

