## [Editor Report]

This work proposes a method to obtain a reduced description of the collective dynamics of thousands of cells moving together during zebrafish gastrulation as a few fundamental modes, and to derive effective dynamics for these modes. This well-written work enables a simplified picture of the key features of cellular collective motion, that will be useful to physicists and biologists looking for a quantitative understanding of morphogenesis.

---

## [Decision Letter]

**Decision letter after peer review:**

Thank you for submitting your article "Learning developmental mode dynamics from single-cell trajectories" for consideration by *eLife*. Your article has been reviewed by 2 peer reviewers, and the evaluation has been overseen by a Reviewing Editor and Naama Barkai as the Senior Editor. The following individual involved in review of your submission has agreed to reveal their identity: Sebastian Fürthauer (Reviewer #2).

Essential revisions:

1) How good is the mode decomposition?

– It is not a priori clear that the mode decomposition suggested is a good representation of the data. It would be helpful if the authors quantified and discussed the error between the continuous low dimensional representation of the data and the raw data.

– Along the same lines, are there a conditions under which mode decomposition is expected to fail? The authors should give a clear idea on when the ideas presented here are applicable and when not.

2) How trustworthy are the inferred summary statistics?

The summary statistic ( defect numbers / fluctuations ) need to be tested against some ground truth. If this is not possible from the data used here it would be illuminating if the authors could test against artificial data.

3) How predictive is the inferred forward model?

– In Figure 3 the authors compare the inferred dynamics and the real data. This is very appealing and the strong point of the paper. We are, however, confused whether the matrix M is inferred using the full time course of the experiment, or just parts of the time course. In other words is M just a way of rewriting the coefficients in Equation 7?

– As it stands we are unclear of what is being learned in M since there is no clear separation between a 'training' and a 'test' set of samples.

4) The paper could greatly benefit if the authors used their method to infer the known dynamics of some artificial data. Then they could validate the inferred model against time courses generated from different initial conditions and against unseen data.

5) We understand that the integral of the field ρ on the surface is 1? We would suggest to mention this already in Equation 1. We also assume that with this definition, J is not actually a cell number flux, but a normalised cell number flux?

6) Because the effective Equations 2a and 2b are written for normalised densities, we understand that cell division and death are effectively absorbed in a flux. So if cells were undergoing cell division in some region of space, and cell apoptosis in another region, this would be interpreted as a net flux between the two regions. Isn't that a potential issue of the method? Can the authors further comment on this?

7) We would argue that one central conclusion of the manuscript is in Figure S6, showing that local rules of interaction cannot explain the dynamics of the system. That is also one potential weakness of the approach: what is exactly learned by the effective dynamics? There is a complex set of interactions between modes suggested by Figure 3D but what is their meaning, given that Figure S6 also indicates, as the authors also point out, that there are missing unresolved players in the equations? The whole approach would be more convincing if the authors could show that the effective learned model have some predictive value. For instance, would changing the initial condition in the integration of the effective system, gives a result that makes sense in regard to the developmental dynamics; indicating that the effective dynamics is not tied to one particular realisation of the biological system?

*Reviewer #1:*

The authors present ideas for obtaining a low dimensional representations of the complex and rich data obtained in cell tracking experiments on zebra fish gastrulation. They argue that such a low dimensional representation can be used to infer a dynamical model from the data. These ideas are potentially very important. However, the methods presented here need to be more explicitly validated.

*Reviewer #2:*

Romeo et al. describe a method for the analysis of the collective flow of cells and its decomposition into modes of motion. The authors start by defining a local density and a local flux by coarse-graining the discrete data in a consistent way. The density and flux fields are then projected onto scalar and vectorial spherical harmonics, and the authors show that a relatively small number of modes is sufficient to account for the flow field; the evolution of some of these modes with lower symmetry reflects major changes in the cell density pattern in the embryo. The authors then look for a linear mode coupling, first order dynamics model for the modes of lowest order they have kept in their analysis. This procedure defines an effective model for the dynamics of the system. Looking in real space at the spatial couplings shows that these effective couplings do not decay spatially, but are long-ranged; indicating that local rules for fluxes cannot capture the dynamics of the system.

The authors presents an elegant and solid work which is done with a very high level of care and rigour. The excellent level of clarity of the manuscript, which presents its concepts in a well-structured manner, makes it really nice to read. The method described by the authors, notably, gives a way to analyse cellular trajectories in a reduced dimensional space which is helpful to characterise the complex cellular flows.

---

## [Author Response]

Essential revisions:1) How good is the mode decomposition?– It is not a priori clear that the mode decomposition suggested is a good representation of the data. It would be helpful if the authors quantified and discussed the error between the continuous low dimensional representation of the data and the raw data.

We thank the reviewer for raising this important point, which has been addressed in detail in the extended Appendix 2 (see also Appendix 2 – Figure 2).

First, we note that it is generally not straightforward to define a meaningful error between the raw particle data, which are restricted to a discrete point set, and the coarse-grained hydrodynamic fields, which are defined on a continuous domain (here given by the surface of the sphere). A theoretically meaningful and practically feasible approach is to quantify the information loss associated with different coarse-graining scales. In the new section of Appendix 2, “Information loss through coarse-graining”, we present measurements of the spectral entropy, which quantifies the variation in information content as the coarse-graining scale is varied. As the coarse-graining scale (kernel width) is increased, one finds that finer structures in the data become rapidly blurred before reaching a slower decay of information (Appendix 2 – Figure 2). Guided by this analysis, we selected a coarse-graining scale (yellow) in the domain where the information loss begins to decay more slowly. This choice ensures that key features of the cell density and fluxes remain well-resolved while also ensuring robustness against variations of the coarse-grain scale.

– Along the same lines, are there a conditions under which mode decomposition is expected to fail? The authors should give a clear idea on when the ideas presented here are applicable and when not.

We thank the reviewer for this comment. As is generally the case for hydrodynamic continuum models, mode decomposition succeeds when gradients in continuous fields are sufficiently smooth. This approach fails when fields are dominated by fine-scale structures or sharp features, such as shock fronts that lead to Gibbs ringing when represented spectrally. In those cases, the mode decomposition has long tails and fails to converge rapidly so that a direct real space representation of data may be more advantageous. In intermediate cases, where convergence is still fast, but the mode representation lacks sparsity, there is no practical benefit in switching from a real-space “pixel basis” to a mode representation. From a computational perspective, a mode representation is advantageous when spectral codes become faster than particle-based or finite-element-based simulation techniques.

For the simulated and experimental systems considered in our study, the density modes decay rapidly with wave-number and the raw flux modes derived from coarse-grained numerical derivatives of noisy data rapidly reach a noise-limited plateau at small mode amplitudes. We have added a supplementary figure to show this more explicitly (Figure 1—figure supplement 1). Therefore, our approach yields a satisfactory reconstruction of the deterministic part of the overall signal in terms of long-wavelength and low-frequency modes that facilitate the inference of hydrodynamic models and are clearly distinguishable from noise. To clarify this further, the revised manuscript contains a newly added Appendix 2 – Figure 1 that complements Video S2 and illustrates that a small number of modes is sufficient to represent the coarse-grained data well. Finally, we note that Appendix 2 provides a detailed description of the threshold-selection procedure (see also Appendix 2 – Figures 3 and 4), where we select thresholding parameters on the Pareto front to optimize the trade-off between reconstruction accuracy and sparsity of the mode representation.

2) How trustworthy are the inferred summary statistics?The summary statistic ( defect numbers / fluctuations ) need to be tested against some ground truth. If this is not possible from the data used here it would be illuminating if the authors could test against artificial data.

We thank the reviewer for this suggestion. We have now added two additional analyses to validate the accuracy and robustness of the defect tracking. First, we successfully tested the defect tracking on simple example harmonic vector modes for which defects can easily be identified by eye (see new Figure 2—figure supplement 2). Second, to verify that experimental defect statistics robustly relate to the underlying microscopic data, we performed the coarse-graining and mode representation with two other kernel length scales and found very similar defect dynamics (new Figure 2—figure supplement 3). We also note that the defects tracked in the experimental data obey at all times the non-trivial condition that defect charges add up to 2, confirming that the extracted information is indeed accurate and robust. Because the defect tracking algorithm works reliably on single snapshots of cell fluxes, dynamic measurements are expected to be similarly robust.

3) How predictive is the inferred forward model?– In Figure 3 the authors compare the inferred dynamics and the real data. This is very appealing and the strong point of the paper. We are, however, confused whether the matrix M is inferred using the full time course of the experiment, or just parts of the time course. In other words is M just a way of rewriting the coefficients in Equation 7?– As it stands we are unclear of what is being learned in M since there is no clear separation between a 'training' and a 'test' set of samples.

We thank the reviewer for this question and have substantially expanded the description of the inference framework (main text pages 10-12 and Appendix 4; see in particular also Appendix 4 – Figure 1 for a detailed flowchart of the learning procedure).

In addition, we now validate the learning framework on synthetic test data generated from the analytically tractable active Brownian particle (ABP) model (pages 10-12 and new Figure 3 in the main text; see also new Appendix 3 for additional details and derivations). For this test case, we follow traditional approaches and divide the data into learning, validation, and prediction segments (new Figure 3D,E). We find that our inferred model fits the learning and validation segments well and has predictive power.

For the experimental data, we divide the data into learning and validation segments. In this case, the linear model Equation 12 (Equation 7) in the initially submitted manuscript with the learned mode coupling matrix *M* can indeed be understood as a compressed representation of the system dynamics. Given the complexity of the experimental data, we require the full range of the data to learn a model that displays good agreement with the training data.

The inferred model can, however, still be used to make predictions: we show in the new Figure 4—figure supplement 1 the output of the model for different initial conditions, such as another experimental zebrafish sample, a rotated initial condition, and an initial condition where 10 percent of the initial density at the animal pole has been numerically “scraped off''. All of these tests show dynamics that reproduce density patterns with the characteristic, elongated “fish shape”, and suggest that indeed the learned model can be used in a predictive context.

4) The paper could greatly benefit if the authors used their method to infer the known dynamics of some artificial data. Then they could validate the inferred model against time courses generated from different initial conditions and against unseen data.

We thank the reviewers for this excellent suggestion. We have now applied our framework to artificial ABP data sets with known ground truth that emulate the key challenges posed by the experimental data. As described in the revised manuscript on pages 10-12 and in Appendix 3, the particle data were generated by simulating the stochastic dynamics of ABPs on a sphere for which an analytic coarse-grained linear continuum model can be derived (Equation 47 in Appendix 3 and Equation 13 in the main text of the revised manuscript). We then applied our inference pipeline consisting of coarse-graining, mode projection, and model inference steps to this particle data. We found that the model inference approach learns a close approximation to the theoretical prediction (new Figure 3), even though the output from the particle data is obscured by noise and spatial information is blurred by coarse-graining. Remarkably, even modes that overlap little with the tested initial condition and thus have a low signal-to-noise ratio were found to follow the model with good agreement.

5) We understand that the integral of the field ρ on the surface is 1? We would suggest to mention this already in Equation 1. We also assume that with this definition, J is not actually a cell number flux, but a normalised cell number flux?

The reviewers are correct that J represents a normalized cell number flux and we thank them for pointing out that this was not explained clearly enough. We have now added this information directly to the text following Equation 1 in line 124 and amended the revised manuscript in line 134.

6) Because the effective Equations 2a and 2b are written for normalised densities, we understand that cell division and death are effectively absorbed in a flux. So if cells were undergoing cell division in some region of space, and cell apoptosis in another region, this would be interpreted as a net flux between the two regions. Isn't that a potential issue of the method? Can the authors further comment on this?

Cell division and death are indeed an important aspect of developmental processes.

As detailed in the main text subsection “Consequences of cell number variations in experimental data”, we show that under an assumption of a spatially-invariant and density-independent – but possibly time-dependent – growth rate, one can absorb effects of varying cell numbers into a normalization of cell density and cell flux. This assumption is motivated by the technical limitation of being unable to distinguish between “true” cell division/death events and imaging or processing artifacts leading to loss of cells, an issue that we discussed with the acknowledged experimental groups that kindly shared and explained their data. As the reviewers correctly point out, inhomogeneous growth would then contribute to effective fluxes. To compensate for those effective inhomogeneous fluxes, we enforce Equation 6 of the main text to relate density and flux modes. This correction only applies to the divergent part of the flux – more details on how we chose spectral projection parameters that optimize reconstruction of flux-derived metrics to heuristically preserve the implicit relationship between divergent and rotational flux components are presented in the section “Optimal compression in space and time” of Appendix 2 (Appendix III B in the previously submitted manuscript).

The overall agreement between the learned model and the experimentally observed dynamics suggests that this procedure provides a good approximation to the exact dynamics of the zebrafish system. We have made limitations arising from currently available data more explicit in the corresponding main text section “Consequences of cell number variations in experimental data”. It is important to emphasize, however, that this is not a fundamental limitation of the mode representation and inference framework. Indeed, as more accurate tracking data is likely to become available in the near future, one can expect that it will soon be possible to distinguish between tracking defects and mitosis or apoptosis (by, say, using suitable mitotic or apoptotic markers). This will make it possible to account for inhomogeneous cell growth in the modeling framework through an extra “growth” field which can be directly included in the local mass balance.

7) We would argue that one central conclusion of the manuscript is in Figure S6, showing that local rules of interaction can not explain the dynamics of the system. That is also one potential weakness of the approach: what is exactly learned by the effective dynamics?

We agree that the learned model in the previous manuscript version while describing the experimental data well, lacked interpretability. To address this point, and guided by the newly added analytical tractable case of ABP dynamics (pages 10-12 and new Figure 3 in the main text), we have substantially expanded the inference framework (new Appendix 4 section “Inference of the dynamical mode coupling matrix *M*” and flow chart Appendix 4 – Figure 1).

The computational improvements, which exploit recent advances in automatic differentiation of ODE solutions, (i) increased the numerical robustness of the inference scheme, (ii) identified a sparser model with higher predictive capability (Figure 4—figure supplement 1), and (iii) enabled through the comparisons with the ABP reference model a direct interpretation of the mode coupling matrix learned for the cellular dynamics during early zebrafish development.

Specifically, the improved model reveals interesting similarities and differences between models of ABPs on the sphere and the dynamics of migrating cells during zebrafish gastrulation. The improved model shows that flux-flux couplings are qualitatively similar for ABPs and zebrafish, while “off-diagonal” density-flux couplings represent longer-range and more inhomogeneous interactions in the experimental data (new Figure 4D, see also new Figure 3C *vs.* new Figure 4A,B). This suggests that one can interpret the experimentally observed collective cell migration in the zebrafish embryos as ABP dynamics with additional interactions.

These new results suggest an intriguing direction for future research: It will be interesting to study through theory and simulations how ABP models with different types of short-, intermediate- and longer-range particles interaction give rise to “off-diagonal” mode-couplings in effective linear models, and if (and how) such models could be improved by including nonlinear coupling terms in Equation 12, similar to triad interactions in nonlinear pattern formation and turbulence models.

There is a complex set of interactions between modes suggested by Figure 3D but what is their meaning, given that Figure S6 also indicates, as the authors also point out, that there are missing unresolved players in the equations?

We agree with the reviewer that Figure 3D, as presented in the previously submitted manuscript, was not ideal for conveying the structure of the learned model. We have therefore replaced this panel with a detailed comparison of the models learned for ABP and zebrafish data, respectively. We believe that, combined with new ABP results in Figure 3, this change leads to much better interpretability of the learned models. As stated in the response to the preceding comment/question, an interesting avenue for future theoretical analysis is to understand how different types of interactions give rise to the “off-diagonal” linear mode-couplings as well as to nonlinear mode couplings in generalized models.

The whole approach would be more convincing if the authors could show that the effective learned model have some predictive value. For instance, would changing the initial condition in the integration of the effective system, gives a result that makes sense in regard to the developmental dynamics; indicating that the effective dynamics is not tied to one particular realisation of the biological system?

We are grateful for this question, which led to the major theoretical and algorithmic improvements included in the revised manuscript. As shown in the new Figure 4—figure supplement 1, the sparser model identified by the enhanced inference framework can be integrated for different initial conditions.

Interestingly, we found that flux and density patterns similar to those seen in the coarse-grained experimental data robustly emerge under various perturbations of the initial conditions. Test examples included in the new Figure 4—figure supplement 1 correspond to an initial condition from the alternative experimental sample, a rotated initial condition, and a removal of cells near the animal pole of the original sample. It would be very interesting to see how the predictions of this model would compare to corresponding cell-displacement experiments in which a subset of cells is ‘surgically’ removed and/or ectopically transplanted during the early stages of zebrafish gastrulation.